# An Embodied Generalist Agent in 3D World

## Abstract

Leveraging massive knowledge and learning schemes from large language models (LLMs), recent machine learning models show notable successes in building generalist agents that exhibit the capability of general-purpose task solving in diverse domains, including natural language processing, computer vision, and robotics. However, a significant challenge remains as these models exhibit limited ability in understanding and interacting with the 3D world. We argue this limitation significantly hinders the current models from performing real-world tasks and further achieving general intelligence. To this end, we introduce an embodied multi-modal and multi-task generalist agent that excels in perceiving, grounding, reasoning, planning, and acting in the 3D world. Our proposed agent, referred to as LEO, is trained with shared LLM-based model architectures, objectives, and weights in two stages: (i) 3D vision-language alignment and (ii) 3D vision-language-action instruction tuning. To facilitate the training, we meticulously curate and generate an extensive dataset comprising object-level and scene-level multi-modal tasks with exceeding scale and complexity, necessitating a deep understanding of and interaction with the 3D world. Through rigorous experiments, we demonstrate LEO's remarkable proficiency across a wide spectrum of tasks, including 3D captioning, question answering, embodied reasoning, embodied navigation, and robotic manipulation. Our ablation results further provide valuable insights for the development of future embodied generalist agents.

## 1 Introduction

Building one generalist model that can achieve comprehensive tasks like humans has been a long-existing pursuit in artificial intelligence and neuroscience (Lake et al., 2015; 2017; Zhu et al., 2020; Mountcastle, 1979; Schmidhuber, 2018; Huang et al., 2022a). Recent advances in large language models (LLMs) (Brown et al., 2020) and "foundation model" (Bommasani et al., 2021) emerge as a promising paradigm in building such generalist models in natural language processing (OpenAI, 2022; 2023), computer vision (Kirillov et al., 2023), and robotics (Brohan et al., 2022; 2023). The keys to the success of this paradigm lie in large-scale internet-level datasets from numerous tasks and domains and scalable Transformer architectures (Vaswani et al., 2017) that can absorb generalizable and task-agnostic knowledge from the data. Such efforts are further extended to multi-modal (Alayrac et al., 2022; Lu et al., 2023; Li et al., 2023c) and generalist models (Reed et al., 2022; Driess et al., 2023) where the agents can solve versatile tasks based on the language-specified task descriptions and show certain generalizability to novel situations. Nonetheless, their abilities are primarily demonstrated within 2D domains, thereby limiting the comprehension of the 3D physical environment that envelops humans and other intelligent species. This limitation acts as an obstacle, preventing current models from successfully executing real-world tasks and the attainment of general intelligence. Therefore, we ask a fundamental question: how to equip the generalist agent with a comprehensive understanding of and the ability to interact with the real 3D world?

The development of such generalist agents encounters three primary challenges: the creation of suitable datasets, the design of unified models, and the design of effective learning strategies. Despite substantial progresses in scaling up image-text models (Tsimpoukelli et al., 2021; Alayrac et al., 2022) and the curation of corresponding datasets (Radford et al., 2021; Schuhmann et al., 2022), advancements in 3D scene-level understanding have significantly lagged behind. This is largely attributed to the limited scale and manual labeling of 3D datasets (Dai et al., 2017; Wald et al., 2019; Chen et al., 2020), given the higher cost associated with collecting 3D data compared to 2D data. Furthermore, previous models have often been designed with strong priors (Zhao et al., 2021; Chen et al., 2022), with limited exploration of large-scale unified pretraining and efficient fine-tuning based on LLMs. Notably, recent works (Zhu et al., 2023c; Hong et al., 2023) utilize the

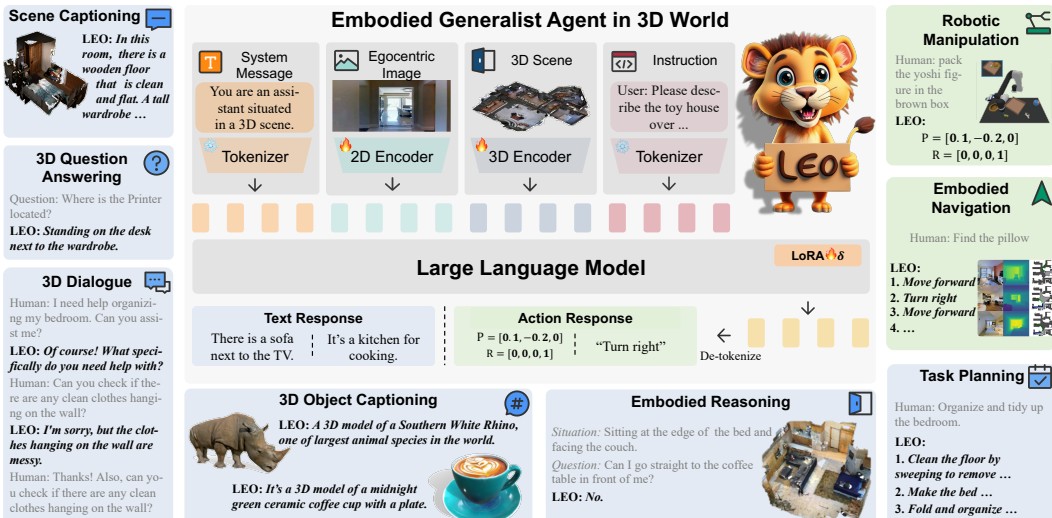

Figure 1: **The proposed embodied generalist agent LEO**. It takes egocentric 2D images, 3D point clouds, and texts as input and formulates comprehensive 3D tasks as autoregressive sequence predictions. By fine-tuning LEO, it extends the capability of LLMs to multi-modal vision-language-action tasks with a unified model.

unified Transformers or LLMs to enhance the model's capability in grounded 3D scene understanding. However, they still lack the agency to act within 3D environments and efforts in unleashing LLMs for 3D vision-language-action (VLA) learning. How to equip the 3D agent with a simple unified architecture and effective learning strategy to establish VLA capability remains rarely explored.

In this work, we introduce the generalist agent LEO, which is generically embodied, multi-modal, and multi-task. It can take egocentric 2D images, 3D point clouds, and texts as task input and achieve comprehensive tasks within the 3D environment. As shown in Fig. 1, LEO *exhibits the capability of perceiving, grounding, reasoning, planning, and acting with shared model architectures and weights.* LEO perceives through an egocentric 2D image encoder for the embodied view and an object-centric 3D point cloud encoder for the third-person global view. The 3D encoder generates an object-centric token for each observed entity. Such encoder design can be flexibly adapted to tasks with various embodiments. These output tokens are then interleaved with text tokens to form a scene-grounded instructional task sequence, which further serves as the input to a decoder-only LLM. All the tasks are re-formulated as sequence prediction problems. Therefore, LEO can be trained with task-agnostic inputs and outputs using autoregressive training objectives. To accommodate embodied tasks that predict action tokens, we employ a pool of special tokens to represent actions and replace the least used tokens (Brohan et al., 2023) in LLM. We perform LoRA (Hu et al., 2022) to fine-tune and adapt the innate knowledge of LLM to the multi-modal generalist. LEO demonstrates two essential capabilities: 3D vision-language grounding and 3D VLA. They are injected with two training stages: 3D vision-language alignment and VLA instruction-tuning. To facilitate the training, we curate a large-scale dataset with object-level and scene-level tasks by fusing existing datasets with high-quality data prompted from the LLMs. We propose scene-graph-based prompting and refinement methods, along with Object-centric Chain-of-Thought (O-CoT) for improving the quality of generated data, largely enriching the data scale and diversity, and further eliminating the hallucination of LLMs.

We quantitatively evaluate and ablate LEO on diverse 3D tasks, including object-level and scene-level captioning (Luo et al., 2023; Chen et al., 2021), 3D question answering (Azuma et al., 2022), situated question answering (Ma et al., 2023), embodied navigation (Ramrakhya et al., 2022), and robotic manipulation (Shridhar et al., 2021). The results indicate (i) LEO achieves state-of-the-art results on most tasks; (ii) through task-agnostic instruction tuning with a unified model, LEO outperforms most previous task-specific models on various domains; (iii) the pretraining of 3D vision-language alignment greatly elevates the performance of VLA instruction-tuning; (iv) scaling up the training data boosts the performance of generalist agent, similar to scaling laws (Kaplan et al., 2020; OpenAI, 2023) in LLMs and generalist agent (Reed et al., 2022). We also show qualitative results of chatting with LEO over diverse tasks, demonstrating its capability in scene-aware planning and dialogue.

In summary, our main contributions are: (i) we propose LEO, the first generalist agent with the capability to perceive, ground, reason, plan, and act in the 3D world; (ii) we demonstrate that a generalist agent can be built via fine-tuning the LLM with object-centric multi-modal representations and mixing the training data with embodied action sequences, enabling it to excel in embodied tasks; (iii) we meticulously curate a large-scale dataset to train such agent and propose several techniques

aimed at enhancing the quality of prompted data from LLMs; (iv) we extensively evaluate LEO and demonstrate its proficiency on diverse tasks including embodied navigation and robotic manipulation, we also observe consistent performance gains while simply scaling up the training data; (v) we will release the data, code, and model weights to endow the future research of generalist agents.

## 2 MODEL

The leading design principles of LEO are two-fold: 1) It should handle the input of egocentric 2D, global 3D, and textual instruction, and the output of textual response as well as embodied action commands in a unified architecture; 2) It should leverage pretrained large language models (LLMs) as a powerful prior for the downstream tasks. We therefore convert all data of different modalities into a sequence of tokens, illustrated below:

$$\underbrace{\texttt{You are...}}_{\text{system message}} \underbrace{s_{2D}^{(1)},...,s_{2D}^{(M)}}_{\substack{\text{2D image tokens} \\ \text{(optional)}}} \underbrace{s_{3D}^{(1)},...,s_{3D}^{(N)}}_{\substack{\text{object-centric} \\ \text{3D tokens}}} \underbrace{\texttt{USER:...} \quad \texttt{ASSISTANT:}}_{\text{instruction}} \underbrace{s_{res}^{(1)},...s_{res}^{(T)}}_{\text{response}}. \quad (1)$$

With this representation, we formulate the learning of LEO as GPT-style autoregressive language modeling (Brown et al., 2020) given the *prefix* (from *system message* to *instruction*), *i.e.* prefix language modeling (Raffel et al., 2020). Therefore, a pretrained LLM can be used to process such sequences. In the following, we will detail the tokenization of multi-modal data, model architecture, training loss, and inference settings. An overview of our model can be found in Fig. 1.

### 2.1 TOKENIZATION

We follow prior practices in 2D VLM (Liu et al., 2023b; Alayrac et al., 2022) and 3D VLM (Zhu et al., 2023c) to tokenize the multi-modal data in LEO. We use SentencePiece tokenizer (Kudo & Richardson, 2018) to encode text with 32k subwords; 2D image tokens for egocentric 2D images; and object-centric 3D tokens extracted over Mask3D-based (Schult et al., 2022) object proposals for 3D point cloud inputs. For embodied action commands, continuous actions (*e.g.* in manipulation) are discretized (details in Appendix B) to join the discrete actions (*e.g.* navigation) and form a unified discrete action space. We follow Brohan et al. (2023) to map these discrete actions to the least used tokens in SentencePiece. After tokenization, all tokens are ordered into the format in (1).

### 2.2 TOKEN EMBEDDING & LLM

We apply several token embedding functions to process the tokens in the sequence before sending them to the LLM. The LLM will then align these tokens of different modalities, and produce the response. Most of the responses are text and can be decoded directly. For responses that include embodied actions, we will map the reserved SentencePiece text tokens back to action commands.

**Text & 2D token embedding.** For text tokens (including embodied actions that have been mapped to the reserved text tokens), an embedding look-up table is used to map them into vectors. While the egocentric 2D image is encoded by a pretrained OpenCLIP ConvNext (Liu et al., 2022) for obtaining image token embeddings. We apply MLP adapters to match the dimensions of all token embeddings.

**Object-centric 3D token embedding.** Each 3D object token (*i.e.*, the point cloud of a 3D object) is first encoded by a pretrained point cloud encoder (*e.g.*, PointNet++ (Qi et al., 2017)). We then adopt the Spatial Transformer introduced in Chen et al. (2022) to further process the point cloud embedding of all objects into object-centric 3D token embeddings. In a nutshell, Spatial Transformer biases the standard attention score with relative position and size for capturing 3D relations between objects. Due to space limit, the readers are referred to Chen et al. (2022) and Appendix D.2 for more details.

**Pretrained LLM.** We choose Vicuna-7B (Chiang et al., 2023) to process the token sequence. In order to tackle the challenging alignment and grounding problem of multi-modal tokens (2D, 3D, text, embodied action) while preserving the LLM pretrained knowledge, we employ LoRA (Hu et al., 2022) to introduce additional tunable parameters to the frozen pretrained LLM.

Table 1: **Datasets statistics**. Top: Datasets for 3D-language alignment; Bottom: Datasets for 3D VLA instruction tuning. *res.* denotes tokens to be predicted in training (response), while *prefix* denotes those in the context. * stands for the tabletop scene setting in CLIPort (Shridhar et al., 2021). † stands for approximate count.

| Dataset | Task | 2D requried? | 3D dataset | #instance | †#token (*res.*) | †#token (*prefix+res.*) |
|---|---|---|---|---|---|---|
| LEO-align | object captioning | ✗ | Objaverse | 660K | 10M | 27M |
|  | object referring | ✗ | ScanNet + 3RScan | 354K | 15M | 39M |
|  | scene captioning | ✗ | 3RScan | 20K | 3.3M | 4.4M |
| LEO-instruct | 3D captioning | ✗ | ScanNet | 37K | 821K | 3M |
|  | 3D QA | ✗ | ScanNet + 3RScan | 83K | 177K | 4M |
|  | 3D dialogue | ✗ | 3RScan | 11K | 1.1M | 8.3M |
|  | task planning | ✗ | 3RScan | 14K | 1.9M | 2.7M |
|  | navigation | ✓ | MP3D | 60K | 11.4M | 272M |
|  | manipulation | ✓ | *CLIPort | 300K | 7.2M | 734M |

## 2.3 TRAINING & INFERENCE

We formulate the learning objective of LEO following (Brown et al., 2020; Raffel et al., 2020) in a prefix language modeling fashion. For a batch $\mathcal{B}$ of token sequence $s$, we optimize LEO via:

$$\mathcal{L}(\theta, \mathcal{B}) = -\sum_{b=1}^{|\mathcal{B}|} \sum_{t=1}^{T} \log p_\theta(s_{\text{res}}^{(b,t)} | s_{\text{res}}^{(b,<t)}, s_{\text{prefix}}^{(b,1)}, ..., s_{\text{prefix}}^{(b,L)}), \tag{2}$$

where $s_{\text{prefix}}$ denotes the prefix token (from *system message* to *instruction*) in (1). During training, we freeze the pretrained 3D point cloud encoder and the LLM and finetune the 2D image encoder, the Spatial Transformer and the LoRA parameters. In total, LEO has ~7B parameters and ~142M of them will be tuned. During inference, we use beam search to generate textual responses. For tasks that require action commands, we map the textual outputs to action commands as discussed in Sec. 2.1. More details on the model and training can be found in Appendix D.

## 3 DATASETS

Since LEO is a generalist agent that receives multi-modal inputs and follows instructions, we adopt the two-stage training proposed in Liu et al. (2023b) and categorize the data into two sets: (i) LEO-align that focuses on **3D vision-language alignment** at object-level and scene-level, to bridge the gap between 3D scene representations and natural language; and (ii) LEO-instruct that targets at **3D VLA instruction tuning** for endowing LEO with the generalist capability of accomplishing myriad tasks in the 3D world including perceiving, reasoning, and acting. We provide the statistics of the two separate splits of data in Tab. 1. Examples of both datasets can be found in Appendix C.

### 3.1 LEO-ALIGN: 3D VISION-LANGUAGE ALIGNMENT

In LEO-align, we focus on 3D vision-language alignment. We follow the alignment method proposed by BLIP-2 (Li et al., 2023d) and train the model to follow the instructions for captioning given 3D input. As a result, we consider collecting the following types of 3D caption data:

**Object-level caption.** To facilitate object-level grounding of detailed object attributes, we leverage Cap3D (Luo et al., 2023), which contains language descriptions for objects in Objaverse (Deitke et al., 2023). Given a single 3D object as input, LEO will be asked to predict its caption.

**Object-in-the-scene caption.** For a better understanding of how an object can be related to others (spatial relations, *etc.*) when situated in a 3D scene, we collect referring expressions of objects in scenes from existing datasets, including ScanScribe (Zhu et al., 2023c) and ReferIt3D (Achlioptas et al., 2020). Further, we generate additional object-referring expressions on 3RScan (Wald et al., 2019) scenes by prompting LLMs (details in Appendix A.1). During alignment, LEO needs to predict these referring expressions given the object-centric 3D input of the scene and the referred object.

**Scene-level caption.** Finally, we encourage LEO to capture scene-level descriptions of a 3D scene. These scene-level captions focus on global information depicting key objects in the scene as well as their attributes and functionalities, relations among multiple objects, and room types and styles. We leverage scene graph annotations (Wald et al., 2019) and prompt LLMs to produce a total of ~20K captions. To further increase caption diversity, we propose a subgraph sampling strategy to prevent

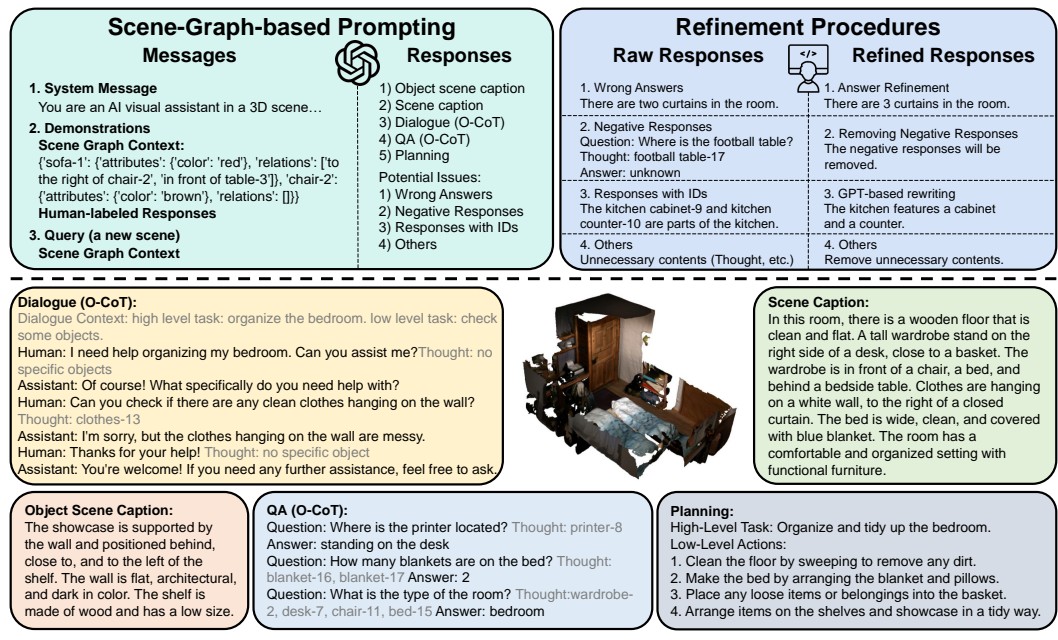

Figure 2: **An overview of our proposed LLM-assisted 3D-language data generation and examples of LEO datasets**. (Top-left) Messages with 3D scene graphs, including object attributes and relations in a phrasal form, used for providing scene context when prompting LLM. (Top-right) The human-defined refinement procedures conducted over raw LLM responses to improve data quality. (Bottom) Examples of LLM-assisted generation in LEO-align and LEO-instruct. Thoughts, colored in gray, will be removed after refinements.

LLMs from always attending to certain notable facets of the scene (details in Appendix A.5). Similar to previous settings, LEO needs to predict these captions given the corresponding 3D input.

## 3.2 LEO-INSTRUCT: INSTRUCTION TUNING FOR TASKS IN THE 3D WORLD

After alignment, LEO will be tuned to follow instructions and accomplish various 3D VLA tasks. Below, we provide a comprehensive illustration of the data preparation process for these tasks and an overview of generated data in Fig. 2. We list the corresponding instructions in Appendix C.

**3D captioning.** The task is to produce a generic caption given 3D input. We adopt the Scan2Cap dataset (Chen et al., 2021), which is based on the ScanNet (Dai et al., 2017) 3D scenes and covers various levels (object-level and scene-level) and aspects (attributes, relations, *etc.*) of scene details.

**3D question answering.** The 3D-QA task is an extension of VQA (Antol et al., 2015) to 3D scenes with a focus on 3D knowledge, ranging from spatial relations to functionalities of objects. For this task, we first aggregate two existing 3D-QA datasets: ScanQA (Azuma et al., 2022) and SQA3D (Ma et al., 2023). To further generate questions concerning rich 3D knowledge, we prompt LLMs to generate ~35K QA pairs on 3RScanQA with our quality refinement techniques discussed in Sec. 3.3.

**3D dialogue.** The goal of this task is to support natural conversations between LEO and users about a given 3D scene. This task necessitates coherence and continuity across multiple rounds of conversational interactions. We build such dialogues on 3RScan scenes by prompting LLMs with a variant of the Chain-of-Thought prompting method discussed in Sec. 3.3 to facilitate diverse dialogues about relevant and accurate details about the 3D scene. In total, ~11K dialogues are collected.

**Scene-aware task planning.** In this task, LEO is required to decompose high-level tasks into step-by-step low-level plans given 3D scenes. We expect LEO to generate feasible plans based on the current 3D scene and ground its inherent common sense knowledge about procedures to the scene configurations, including, objects, their attributes, relations, and functional characteristics, *etc*. By prompting LLMs, we end up collecting ~14K task-plan pairs on 3RScan scenes.

**Embodied navigation.** We follow imitation learning setting in Habitat-web (Ramrakhya et al., 2022) for the embodied navigation task. We choose `ObjNav`, where LEO needs to map navigation instructions (*e.g.* "find bed"), object-centric 3D input, and an egocentric 2D input into discrete habitat motor commands. For simplicity, we use shortest path navigation trials rather than human demonstrations for learning as they are less noisy and therefore easier to learn when provided with

the 3D scene. In total, we generate ~60K navigation episodes out of the MP3D `ObjNav` training scenes (Savva et al., 2019) for this task.

**Robotic manipulation.** We employ a subset of the manipulation tasks introduced in CLIPort (Shridhar et al., 2021). The input of this task includes instructions, egocentric 2D observations, and object-centric 3D information. As discussed in Sec. 2.1, we discretize the continuous action space of CLIPort into bins to unify the action decoding of navigation and manipulation (more details in Appendix B). We generate 100K demonstrations for each selected manipulation task.

### 3.3 LLM-assisted 3D-language Data Generation

As mentioned above, at the core of producing a large proportion of LEO-align and LEO-instruct is the assistance of LLMs. We now detail the key techniques of prompting LLMs (more specifically, ChatGPT) to generate 3D-text data. An overview can be found in Fig. 2.

**Scene-graph-based prompting.** We use 3D scene graph from 3DSSG (Wu et al., 2021) to provide scene contexts in prompts. Compared to recent efforts that utilize object boxes (Yin et al., 2023; Hong et al., 2023; Wang et al., 2023d), we observed that our method provides high-quality object attributes and spatial relation information among objects, allowing LLMs to capture and generate more accurate and relevant 3D details (see comparisons in Appendix A.6). To further improve data quality in open-ended generation and reduce the hallucination of LLMs (Bang et al., 2023), we propose the Object-centric chain of thought (O-CoT) prompting that requires the LLM to explicitly provide the label and ID of object candidates as thoughts during question and dialogue generation. We provide examples of O-CoT in Fig. 2 and comparative experiments to verify the effectiveness of O-CoT on improving answer reliability in Appendix A.2. We also utilize subgraph sampling to further enhance the diversity of 3D scene graph (see details in Appendix A.5).

**Refinement procedures.** We pass raw LLM-generated responses into several human-defined filtering procedures based on the 3D scene graph. Notably, negative responses (*e.g.*, lacking necessary information to answer) will be removed; unnatural narratives will be rewritten. For generated text that involves logical reasoning (*e.g.*, counting) or hallucination, we manually fix the wrong responses based on the information provided by the 3D scene graph. We provide details about these procedures in Appendix A.3 and statistics in Appendix A.4.

## 4 Capabilities and Analyses

We present a comprehensive demonstration of LEO's capabilities by evaluating it on the full spectrum of embodied 3D tasks encompassing perceiving, grounding, reasoning, planning, and acting. We provide both quantitative comparisons between LEO and competitive task-specific baselines and qualitative visualizations (see in Fig. 3) to showcase the power of LEO as an embodied generalist agent. We provide additional experimental details about the model and implementation in Appendix D. We further ablate LEO with various data configurations and analyze the scaling law.

### 4.1 3D Vision-Language Understanding and Embodied Reasoning

**Overview.** Understanding and reasoning about object attributes, object relations, and other facets of 3D scenes from an agent's egocentric perspective is a fundamental capability of an embodied generalist agent in the 3D world. We investigate how well can LEO perform 3D VL understanding and embodied reasoning tasks, especially when being compared against task-specific models and existing generalist agents. Specifically, we consider three renowned 3D tasks: 3D captioning on Scan2Cap (Chen et al., 2021), 3D QA on ScanQA (Azuma et al., 2022), and 3D embodied reasoning on SQA3D (Ma et al., 2023). By prompting LEO to follow instructions for these tasks, we follow the standard evaluation metric to report conventional captioning scores (CIDEr, BLEU, METEOR, and ROUGE) and SentenceSim (Reimers & Gurevych, 2019) for open-ended VL generation, as well as exact-match accuracies for QA tasks. Following 3D-VisTA (Zhu et al., 2023c), we use object proposals from Mask3D (Schult et al., 2022) in our object-centric 3D encoder.

**Baselines.** For quantitative comparisons, we include both task-specific approaches and generalist models: 1) state-of-the-art specialists in 3D dense captioning (Chen et al., 2021; Cai et al., 2022; Chen et al., 2023); 2) state-of-the-art specialists in 3D QA (Azuma et al., 2022; Ma et al., 2023); 3) task-specific fine-tuned generalist models like 3D-VisTA (Zhu et al., 2023c) and 3D-LLM (Hong et al.,

Table 2: **Quantitative comparison with state-of-the-art models on 3D VL understanding and embodied reasoning tasks**. "C" stands for "CIDEr", "B-4" for "BLEU-4", "M" for "METEOR", "R" for "ROUGE", "Sim" for sentence similarity, and "EM@1" for top-1 exact match. The n-gram metrics for Scan2Cap are governed by IoU@0.5. [†] indicates answering questions via prompting GPT-3 with the generated scene caption.

| | Scan2Cap (val) | | | | | ScanQA (val) | | | | | SQA3D |
|---|---|---|---|---|---|---|---|---|---|---|---|
| | C | B-4 | M | R | Sim | C | B-4 | M | R | EM@1 | EM@1 |
| *Task-specific models* | | | | | | | | | | | |
| Scan2Cap (GPT-3) (Chen et al., 2021) | 35.2 | 22.4 | 21.4 | 43.5 | - | - | - | - | - | - | 41.0[†] |
| 3DJCG (Cai et al., 2022) | 47.7 | 31.5 | 24.3 | 51.8 | - | - | - | - | - | - | - |
| Vote2Cap-DETR (Chen et al., 2023) | 61.8 | 34.5 | 26.2 | 54.4 | - | - | - | - | - | - | - |
| ScanRefer+MCAN (Chen et al., 2020) | - | - | - | - | - | 55.4 | 7.9 | 11.5 | 30.0 | 18.6 | - |
| ClipBERT (Lei et al., 2021) | - | - | - | - | - | - | - | - | - | - | 43.3 |
| ScanQA (Azuma et al., 2022) | - | - | - | - | - | 64.9 | 10.1 | 13.1 | 33.3 | 21.1 | 47.2 |
| *Task-specific fine-tuned* | | | | | | | | | | | |
| 3D-VisTA (Zhu et al., 2023c) | 66.9 | 34.0 | 27.1 | 54.3 | 53.8 | 69.6 | 10.4 | 13.9 | 35.7 | 22.4 | 48.5 |
| 3D-LLM (FlanT5) (Hong et al., 2023) | - | - | - | - | - | 69.4 | **12.0** | 14.5 | 35.7 | 20.5 | - |
| LEO | **68.4** | **36.9** | **27.7** | **57.8** | **54.7** | **80.0** | 11.5 | **16.2** | **39.3** | **36.6** | **53.7** |

Table 3: **Results on CLIPort robot manipulation**. We compare with results from Shridhar et al. (2021). seen indicates in-domain tasks. unseen marks OOD tasks with novel colors or objects.

| | separating-piles | | packing-google -objects-seq | | put-blocks-in -bowls | |
|---|---|---|---|---|---|---|
| | seen | unseen | seen | unseen | seen | unseen |
| Transporter | 48.4 | 52.3 | 46.3 | 37.3 | 64.7 | 18.7 |
| CLIP-only | 90.2 | 71.0 | 95.8 | 57.8 | 97.7 | 44.5 |
| RN50-BERT | 46.5 | 44.9 | 94.0 | 56.1 | 91.8 | 23.8 |
| CLIPort (single) | 98.0 | **75.2** | **96.2** | 71.9 | **100** | 25.0 |
| CLIPort (multi) | 89.0 | 62.8 | 84.4 | 70.3 | **100** | **45.8** |
| LEO | **98.8** | **75.2** | 76.6 | **79.8** | 86.2 | 35.2 |

Table 4: **Results on object navigation.** We compare LEO with similar imitation learning agents in Habitat-web (H.w.) (Ramrakhya et al., 2022) and CortexBench (VC-1) (Majumdar et al., 2023). S: success rate; L: SPL. [†] LEO is not trained on HM3D scenes.

| | MP3D-val | | HM3D-val | |
|---|---|---|---|---|
| | S(↑) | L(↑) | S(↑) | L(↑) |
| H.w. (shortest) | 4.4 | 2.2 | - | - |
| H.w. (70k demo) | **35.4** | 10.2 | - | - |
| VC-1 (ViT-B) | - | - | **57.1** | **31.4** |
| LEO | 23.1 | **15.2** | 23.1[†] | 19.1[†] |

2023). To the best of our knowledge, LEO is the first model that, in stark contrast to prior models, can handle the aforementioned VL tasks in a unified architecture without additional fine-tuning. This lends greater credence to LEO's comparative superiority.

**Results & analysis.** As shown in Tab. 2, LEO surpasses state-of-the-art task-specific and task-specific fine-tuned models significantly on both 3D dense captioning and 3D QA tasks. In contrast to the specialist models that utilize task-specific heads, we demonstrate our LLM-based approach **not only affords the flexibility of generating open-ended responses but also can achieve excellent scores in terms of standard metrics**. Notably, considering the difference between close-set classification (*e.g.*, 3D-VisTA) and open-ended text generation, we refine the protocol of exact match (see details in Appendix H.1). On the other hand, compared with the complicated 3D feature aggregation in 3D-LLM, we suggest that **object-centric 3D representation is a simple yet effective option to connect 3D scenes with LLM while harnessing the inherent knowledge of LLM**.

## 4.2 CHATTING AND PLANNING ABOUT A 3D SCENE

**Overview.** Upon the 3D VL understanding and reasoning, we anticipate LEO to support more sophisticated and grounded interaction with human users, *i.e.*, responding to complex multi-round user instructions in the 3D world. To verify these capabilities, we choose two tasks: 3D dialogue and scene-aware task planning. We provide qualitative examples of unseen scenarios from the held-out test sets of LEO-instruct, highlighting LEO's merits of instruction following and scene-grounded responses. We defer the quantitative results of dialogue and planning to our ablation study in Sec. 4.4. Quantitative comparison with other approaches is infeasible due to the lack of a common benchmark.

**Results & analysis.** As shown in Fig. 3, LEO is capable of generating high-quality responses to complete the tasks of dialogue and planning. We highlight two features: **1) The responses of LEO are precisely grounded to the 3D scenes.** In particular, the proposed plan by LEO contains concrete objects that are present in the scenes, as well as concrete actions regarding these objects. **2) The responses of LEO incorporates rich informative spatial relations.** Such information is necessary to refer to specific objects in complex 3D scenes and affords considerable assistance for humans.

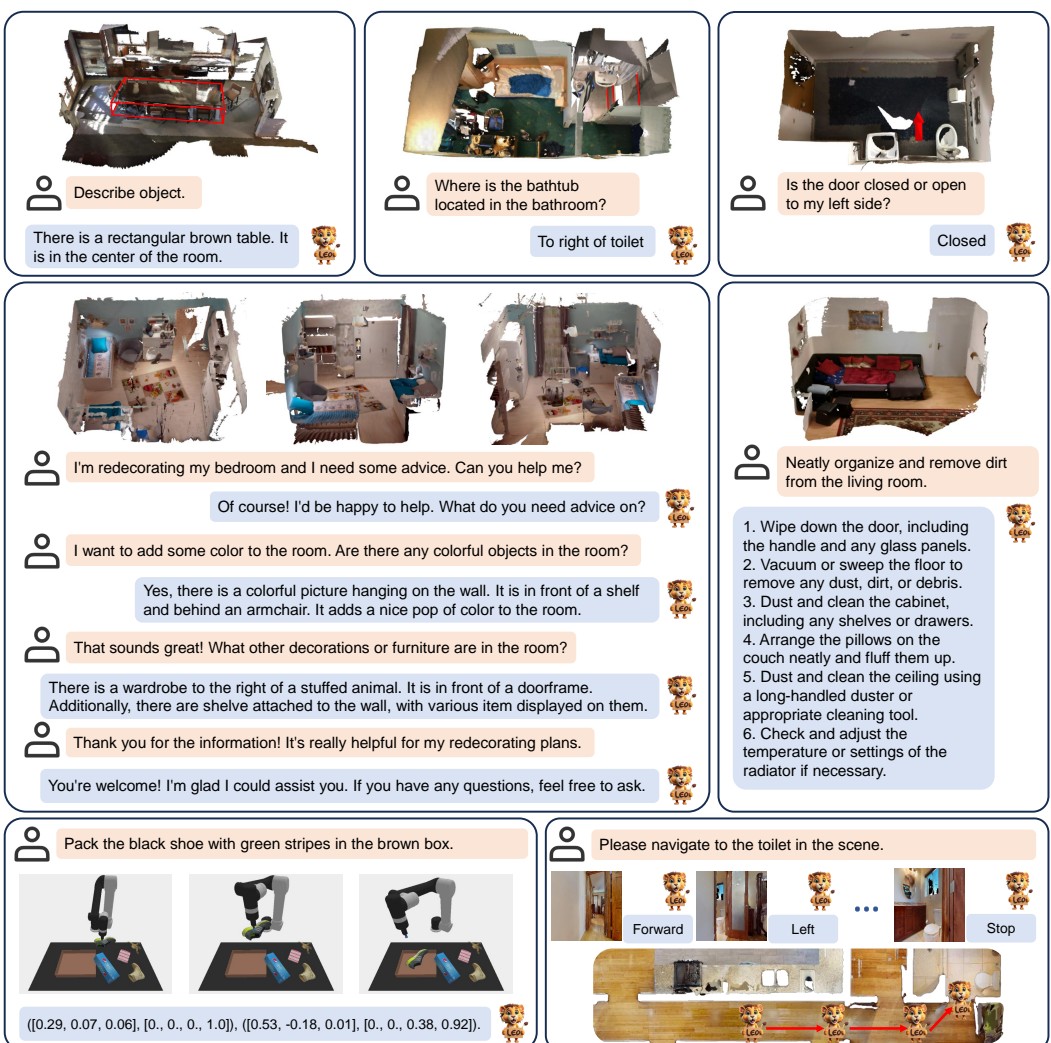

Figure 3: **Qualitative results of interacting with LEO** on unseen scenarios from a held-out test set of LEO-instruct. LEO's responses and actions can be grounded in novel scenes.

## 4.3 EMBODIED ACTION IN 3D WORLD

**Overview.**    Finally, we hope to directly probe the embodied acting and interacting capacity of LEO in the 3D World. We select two canonical embodied AI tasks: embodied navigation with `ObjNav` on AI Habitat (Ramrakhya et al., 2022) and robotic manipulation on CLIPort (Shridhar et al., 2021). Specifically, for `ObjNav`, although LEO is trained on a customized dataset (see Sec. 3.2), the scenes are all included in the original MP3D `ObjNav` training split (Savva et al., 2019). Therefore, we still evaluate LEO on the original MP3D `ObjNav` validation split against baselines. Additionally, we test LEO on the validation split of the newly introduced HM3D `ObjNav` task (Ramakrishnan et al., 2021). We report the success rate and SPL metrics following Ramrakhya et al. (2022). For CLIPort robotic manipulation, we evaluate LEO on the three training tasks listed in Tab. 3 and their corresponding unseen tasks and report the average reward across the evaluation episodes.

**Results & analysis.**    We present the results of CLIPort manipulation and object navigation in Tabs. 3 and 4. Our findings are as follows: 1) In robotic manipulation, LEO exhibits comparable performances to many strong baselines and even achieves significantly better results on some challenging unseen tasks. Note that compared to baselines that rely on heatmap output, LEO produces motor commands directly. 2) On `ObjNav`, LEO attains a reasonable success rate and better SPL on MP3D-val compared with baselines, suggesting that LEO can learn to leverage the object-centric 3D scene input (potentially offering a coarse global map) and take a shorter path to the target. Further, results

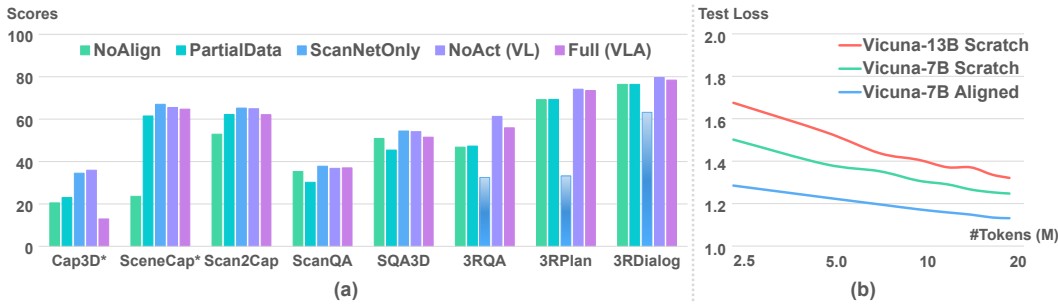

Figure 4: (a) Comparison of LEO trained under different data configurations. We test models on the validation set of Scan2Cap and ScanQA, and the held-out test sets of other datasets. * indicates datasets that belong to LEO-align. The color-gradient bar of *ScanNetOnly* on 3RQA, 3RPlan and 3RDialog indicates zero-shot transfer. Refer to Appendix G for numerical results. (b) Instruction-tuning losses on the test set, with the growth of data and model scale. These results echo the scaling law introduced in Kaplan et al. (2020); Reed et al. (2022).

on HM3D-val confirm LEO's zero-shot generalization to novel scenes. Please note that all baselines use an RNN policy while LEO can be viewed as a transformer-based feed-forward policy similar to RT-2 (Brohan et al., 2023) considering the training efficiency, which could lead to a lower success rate. More discussions on this can be found in Appendix H.2. 3) Overall, the align-then-instruct tuning scheme endows LEO with semantic-level generalization (novel objects, *etc.*) in both manipulation and navigation tasks (we provide results on `ObjNav` with unseen objects in Appendix I.1).

## 4.4 ABLATIVE STUDY

**Settings.** We ablate LEO on different data configurations. Specifically, we compare LEO's performance under the following settings: (1) *NoAlign*: tuning LEO from scratch on LEO-instruct, skipping pre-training on LEO-align; (2) *PartialData*: uniformly sampling 10% of data in LEO-instruct during instruction-tuning; (3) *ScanNetOnly*: excluding data generated by LLM and embodied tasks (*i.e.*, navigation and manipulation) during instruction tuning; (4) *NoAct (VL)*: excluding embodied task data during instruction tuning. We provide additional results and findings in Appendix G.

**Evaluation.** We provide a more comprehensive quantitative evaluation of LEO on all 3D VL tasks, including 3D object/scene captioning, 3D QA, scene-grounded dialogue and task planning. Following prior works (Achlioptas et al., 2020), we use ground-truth object proposals in our ablation to pinpoint the reasoning and planning capabilities of LEO. We report exact match scores for QA tasks and SentenceSim for all other tasks. Fig. 4 shows a holistic view of the results.

**Results & analysis.** **1) The two-stage align-then-instruct pipeline is critical for LEO learning.** The lack of alignment harms detailed understanding of scenes, while the decrease in instruction-tuning data affects reasoning and planning. **2) Compositional generalization poses considerable challenges.** *ScanNetOnly*, having been exposed to 3RScan scenes or QA skills during the two stages respectively, still struggles to handle the QA task in 3RScan scenes (3RQA). **3) General vs. specific.** We observe model performance drops on in-domain tasks (*e.g.*, Scan2Cap) when adding data from other domains or new tasks (*ScanNetOnly vs. NoAct (VL)*). Scaling up the instruction-tuning data brings significant improvements, though the embodied acting data counteracts such effects due to the domain gap (*PartialData vs. Full (VLA) vs. NoAct (VL)*).

## 4.5 SCALING LAW ANALYSIS

**Settings.** Following the analysis in Sec. 4.4, we study the scaling effect (Kaplan et al., 2020; Reed et al., 2022) of data and model in LEO. We use the instruction-tuning loss (on the test set) of LEO with the growth of data and model scale as an indicator. Based on *NoAct (VL)* with Vicuna-7B (referred to as *Vicuna-7B Aligned*), we add two variants: (1) *Vicuna-7B Scratch*, trained without the alignment stage; and (2) *Vicuna-13B Scratch*, trained without the alignment stage and scaling up the LLM to 13B. The curves of test loss are visualized in Fig. 4(b).

**Results & analysis.** **1) The instruction tuning of LEO conforms to the scaling law** (Kaplan et al., 2020; Reed et al., 2022). For all three settings, we find the test loss of LEO decreases log-linearly as it is fed with more data. **2) The lack of alignment causes significantly higher loss.** *Vicuna-7B Scratch* shows consistently higher loss than *Vicuna-7B Aligned*. This corresponds to the inferior performances of *NoAlign* in Sec. 4.4 and emphasizes the importance of alignment. **3) Scaling up LLM leads to degradation.** *Vicuna-13B Scratch* shows consistently higher loss than *Vicuna-7B*

*Scratch*, which echos previous findings (Dai et al., 2023; Xu et al., 2023). We conjecture there are two possible reasons: multi-modal instruction tuning data is insufficient to reveal the benefit of scaling up LLMs, or a small-scale LLM (*e.g.*, Vicuna-7B) already suffices for connecting the visual modality.

## 5 RELATED WORK

**Generalist agents.** The AI community has witnessed the rising generalist models in both vision (Lu et al., 2023; Wang et al., 2023b; Kirillov et al., 2023) and language (OpenAI, 2022; 2023) domains. A generalist agent requires additional embodiment knowledge to interact with the environment and complete embodied acting tasks. Existing efforts towards generalist agents include: grounded reasoning and task planning in the real world (Ahn et al., 2022; Huang et al., 2022b), skill generalization in open-world environment (Fan et al., 2022; Cai et al., 2023a; Wang et al., 2023e;a; Cai et al., 2023b; Gong et al., 2023b), general robotic manipulation (Brohan et al., 2022; Jiang et al., 2023; Gong et al., 2023a), and unified vision-language-action (VLA) models such as Gato (Reed et al., 2022), PaLM-E (Driess et al., 2023), EmbodiedGPT (Mu et al., 2023), and RT-2 (Brohan et al., 2023). LEO belongs to the VLA model, however, its goal is to build a generalist agent that can understand the real 3D world beyond 2D images, which is absent in existing works.

**Multi-modal instruction tuning.** Pre-trained LLMs demonstrated practical for solving vision-language tasks (Tsimpoukelli et al., 2021; Alayrac et al., 2022; Guo et al., 2023; Li et al., 2023d; Zhao et al., 2023). Meanwhile, the instruction-tuning paradigm exhibited strong zero-shot generalization in NLP tasks (Wei et al., 2022; Sanh et al., 2022; Ouyang et al., 2022; Chung et al., 2022). The two streams merged into instruction-tuned LVLMs (Liu et al., 2023b; Zhu et al., 2023b; Ye et al., 2023; Gao et al., 2023; Li et al., 2023b; Gong et al., 2023c; Dai et al., 2023). Despite the burst, these models are confined to 2D visual modalities, *e.g.*, image or video. Concurrent works (Yin et al., 2023; Hong et al., 2023; Wang et al., 2023d; Xu et al., 2023) extend to 3D vision tasks, but these models either lack the acting capability or unified efficient architecture.

**Grounded 3D scene understanding.** One key obstacle to building LEO is grounding the 3D world with natural languages. There exist diverse methods of grounded scene understanding, *e.g.*, spatial relation modeling (Zhao et al., 2021; Chen et al., 2022; Zhu et al., 2023c) and fine-grained open-scene understanding (Peng et al., 2023b; Kerr et al., 2023). However, due to data scarcity, how to utilize LLMs to ground the 3D scene is rarely explored. Recently, 3D-LLM (Hong et al., 2023) leverages multi-view images and Chat-3D (Wang et al., 2023d) uses object-centric point clouds to enable the LLMs with 3D grounding. In this work, we devise both 2D and 3D encoders for grounding various visual representations and employ LoRA (Hu et al., 2022) to efficiently fine-tune the LLMs.

**3D data prompting from LLMs.** LLMs exhibit extraordinary capabilities of text generation and serve as a source for collecting diverse instruction-following data (Wang et al., 2023c; Taori et al., 2023; Peng et al., 2023a). However, the lack of access to visual modalities makes it troublesome to collect visual instruction-tuning data. To address this issue, existing methods provide bounding boxes (Liu et al., 2023b) and add dense captions (Li et al., 2023a; Liu et al., 2023a) as image descriptions or directly use off-the-shelf large vision-language models (LVLM) (Zhu et al., 2023a; Luo et al., 2023) to help collect such data. Unlike concurrent attempts (Yin et al., 2023; Hong et al., 2023; Wang et al., 2023d) in collecting 3D instruction-tuning data, our approach features a scene-graph-based prompting and refinement method to prompt and correct the data.

## 6 CONCLUSIONS

The proposed agent LEO extends the current generalist ability of LLMs from text towards the 3D world and embodied tasks. It is a crucial initial step toward building embodied generalist agents. In light of this work, we identify several promising directions that hold the potential for substantial advancement: (1) enhancing the 3D vision-language grounding capability by leveraging larger-scale paired data from richer real-world 3D domains; (2) continually bridging the gap between 3D vision-language and embodied action, as our experiments reveal the feasibility of their joint learning; (3) investigating the issues of safety and alignment in the context of embodied generalist agents, particularly given that our scaling law analysis suggests that such agents can experience significant enhancements through data scaling in the near future.

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

# A DATA

## A.1 PROMPTS FOR LLM-ASSISTED 3D DATA GENERATION

In Fig. 5–9, we show the prompts for five types of LLM-assisted 3D-language data generation. We provide few-shot examples as the context. In each example, the "content" contains a scene graph, and the "response" refers to a human-labeled response. The query is a new scene graph, based on which ChatGPT (OpenAI, 2022) generates responses.

```
messages = [{''role'': ''system'', ''content'': ''You are an AI visual assistant in a 3D scene. The scene
contains some objects, which compose a scene graph in json format. Each entity in the scene graph denotes an object instance,
with a class label and an object id. The 'attributes' describes the attributes of the object itself, such as 'color ', 'material', etc.
The 'relations' describes the spatial relations with other objects.
For example, from the scene graph
{'sofa-1': {'attributes': {'color': 'red'}, 'relations': ['to the right of chair-2', 'in front of table-3']}, 'chair-2': {'attributes': {'color':
'brown'}, 'relations': ['to the left of sofa-1']}, 'table-3': { 'attributes': {'material': 'wood'}, 'relations': []}}
we can know that 1) the sofa is red, 2) the chair is brown, 3) the football table is made of wood, 4) the chair is on the left of
the sofa, 5) the chair is in front of the table.
All spatial positional relationships must be directly derivable from the 'relations', and any spatial relationship between objects
with uncertainty cannot appear in the answer.

You need to generate meaningful conversations based on the scene information. The conversations include questions
from human and responses from an AI assistant. Ask questions about the object types, counting the objects, object attributes,
relative positions between objects. Also ask questions concerning commonsense, e.g., how the objects can be used by human
and human activity in the scene. You can ask questions about the affordance of the objects in the scene. The questions should
conform to the given scene information. The attributes of objects and spatial relations between objects can only be inferred
from the 'attributes' and 'relations' in scene graph, respectively. The questions should contain interrogative sentences and
declarative sentences to cover diverse tones. You need to first provide the context of the dialogue. The context can be high
level or low level tasks. The dialogue should be related to the context. Then you need to provide the clues about the question.
Then the robot answers the question according to the thought. The dialogue has the following format:Dialogue Context:
<Dialogue Context>\nHuman:<Question>\nThought:<Thought>\nRobot:<Answer>. Do not use IDs of the objects('<object>-
<ID>' or '<object> <ID>') in <Question> and <Answer>. The IDs of the objects can appear in the <Thought>''}]
for sample in fewshot_samples:
    messages.append({''role'': ''user'', ''content'': sample['content']})
    messages.append({''role'': ''assistant'', ''content'': sample['response']})
messages.append ({''role'': ''user'', ''content'': '\n'.join(sample['query'])})
```

Figure 5: The prompt for generating 3D Dialogue.

```
messages = [{''role'': ''system'', ''content'': ''You are an AI visual assistant in a 3D scene. The scene
contains some objects, which compose a scene graph in json format. Each entity in the scene graph denotes an object instance,
with a class label and an object id. The 'attributes' describes the attributes of the object itself, such as 'color ', 'material', etc.
The 'relations' describes the spatial relations with other objects.
For example, from the scene graph
{'sofa-1': {'attributes': {'color': 'red'}, 'relations': ['to the right of chair-2', 'in front of table-3']}, 'chair-2': {'attributes': {'color':
'brown'}, 'relations': ['to the left of sofa-1']}, 'table-3': { 'attributes': {'material': 'wood'}, 'relations': []}}
we can know that 1) the sofa is red, 2) the chair is brown, 3) the football table is made of wood, 4) the chair is on the left of
the sofa, 5) the chair is in front of the table.
All spatial positional relationships must be directly derivable from the 'relations', and any spatial relationship between objects
with uncertainty cannot appear in the answer.

You need to generate 10-15 question-answer pairs based on the scene information. The question-answer pairs include the
object types, counting the objects, object attributes, relative positions between objects. The questions should conform to the
given scene information. The attributes of objects and spatial relations between objects can only be inferred from the
'attributes' and 'relations' in scene graph, respectively. The questions must be able to be answered correctly based on the scene
graph. You need to provide the queried object. Note that all answers to the questions must be single words or phrases. The
question answer pair should be following format:\nQ: <question>\nT: <queried object(s)>\nA: <Answer>. You can answer
the question according to the queried object(s). If there is no information about the question, the <Answer> should be
'unkown'.''}]
for sample in few_shot_samples:
    messages.append({''role'': ''user'', ''content'': sample['content']})
    messages.append({''role'': ''assistant'', ''content'': sample['response']})
messages.append ({''role'': ''user'', ''content'': '\n'.join(sample['query'])})
```

Figure 6: The prompt for generating 3D QA.

Fig. 5 shows the prompt for generating 3D dialogue data. Red fonts outline our requirements of the dialogue content, including object attributes, spatial relations, and commonsense topics. Purple fonts formulate the template of the response. We require the response generated by the ChatGPT should include the dialogue context as well; the "thought" contains the involved objects in the question, which is used to enhance the reliability of the answer. These two components will be removed after the refinement procedures.

```
messages = [{''role'': ''system'', ''content'': ''You are an AI visual assistant that can analyze a 3D
scene. The scene contains some objects, which compose a scene graph in json format. Each entity in the scene graph denotes
an object instance, with a class label and an object id. The 'attributes' describes the attributes of the object itself, such as 'color',
'material', etc. The 'relations' describes the spatial relations with other objects.
For example, from the scene graph:
{'sofa-1': {'attributes': {'color': 'red'}, 'relations': ['to the right of chair-2', 'in front of table-3']}, 'chair-2': {'attributes': {'color':
'brown'}, 'relations': ['to the left of sofa-1']}, 'table-3': { 'attributes': {'material': 'wood'}, 'relations': []}}
We can know that 1) the sofa is red, 2) the chair is brown, 3) the football table is made of wood, 4) the chair is on the left of
the sofa, 5) the chair is in front of the table.
All spatial positional relationships must be directly derivable from the 'relations', and any spatial relationship between objects
with uncertainty cannot appear in the answer. Do not use the id of the object in the dialogue, use ordinal words and attributes
to refer to different objects with the same label.

Using the provided scene graph, design a high-level task that can be performed in this 3D scene. Besides, decomposing
this high-level task into a sequence of action steps that can be performed using the instances in this3D scene.
Remeber, the high-level task and action steps must be able to be performed in the 3D scene using the given object instances.
Do not use IDs of the objects('<object>-<ID>' or '<object> <ID>') in the planning.''}]
for sample in fewshot_samples:
    messages.append({''role'': ''user'', ''content'': sample['content']})
    messages.append({''role'': ''assistant'', ''content'': sample['response']})
messages.append ({''role'': ''user'', ''content'': '\n'.join(sample['query'])})
```

Figure 7: The prompt for generating 3D planning.

```
messages = [{''role'': ''system'', ''content'': ''You are an AI visual assistant in a 3D scene. The scene
contains some objects, which compose a scene graph in json format. Each entity in the scene graph denotes an object instance,
with a class label and an object id. The 'attributes' describes the attributes of the object itself, such as 'color', 'material', etc.
The 'relations' describes the spatial relations with other objects.
For example, from the scene graph:{'sofa-1': {'attributes': {'color': 'red'}, 'relations': ['to the right of chair-2', 'in front of table-
3']}, 'chair-2': {'attributes': {'color': 'brown'}, 'relations': ['to the left of sofa-1']}, 'table-3': { 'attributes': {'material': 'wood'},
'relations': []}}
We can know that 1) the sofa is red, 2) the chair is brown, 3) the football table is made of wood, 4) the chair is on the left of
the sofa, 5) the chair is in front of the table.
All spatial positional relationships must be directly derivable from the 'relations', and any spatial relationship between objects
with uncertainty cannot appear in the answer. Don't use IDs of the objects('<object label>-<ID>' or '<object label> <ID>') in
the summary.

You need to provide a summary for a scene. The summary should be about the object types, object attributes, relative
positions between objects. Also describe the scene concerning commonsense, e.g., how the objects can be used by human and
human activity in the scene. The description should conform to the given scene information. The attributes of objects and
spatial relations between objects can only be inferred from the 'attributes' and 'relations' in scene graph, respectively. You
don't need to describe each object in the scene, pick some objects of the scene for summary. You can also summarize the
room's function, style, and comfort level based on the arrangement and color of objects within the room. Your summary must
not exceed 110 words.''}]
for sample in few_shot_samples:
    messages.append({''role'': ''user'', ''content'': sample['content']})
    messages.append({''role'': ''assistant'', ''content'': sample['response']})
messages.append ({''role'': ''user'', ''content'': '\n'.join(sample['query'])})
```

Figure 8: The prompt for generating 3D scene caption.

## A.2 ANALYSIS OF THE OBJECT-CENTRIC CHAIN-OF-THOUGHT

To further investigate the impact of Object-centric Chain-of-Thought (O-CoT) on data quality, we analyze the answer accuracy for Object Counting questions. Specifically, we collect several demonstrations, and for each run, we select two of them as the prompt seed. With these seeds, we generate dialogues across all scenes in 3DSSG (Wu et al., 2021) and then assess the answer accuracy for Object Counting questions. The results are presented in Tab. 5.

```
messages = [{''role'': ''system'', ''content'': ''You are a helpful assistant. You will receive a dictionary
of an object. This dictionary provides information about a node in a scene graph, as well as its adjacent nodes. The value of
the key 'object' is the object represented by the node. The value of 'relations' includes the spatial relationships with the
adjacent nodes . The value of the key 'attribute' provides the attributes of the object. The value of 'edge attribute' provides a list
of object attributes for the adjacent nodes. You need to describe the object according to the information of the target
object node. The IDs of objects cannot appear in the summary.''}]
for sample in few_shot_samples:
    messages.append({''role'': ''user'', ''content'': sample['content']})
    messages.append({''role'': ''assistant'', ''content'': sample['response']})
messages.append ({''role'': ''user'', ''content'': '\n'.join(sample['query'])})
```

Figure 9: The prompt for generating 3D object-in-the-scene caption.

Table 5: The effect of O-CoT on the answer accuracy for Object Counting questions.

| Settings | Seed 1 | Seed 2 | Seed 3 | Seed 4 | Average | Avg. Gain |
|----------|--------|--------|--------|--------|---------|-----------|
| w/o O-CoT | 0.5838 | 0.5349 | 0.5962 | 0.5816 | 0.5741 | 0.2061 |
| O-CoT | 0.7647 | 0.8117 | 0.7778 | 0.7667 | 0.7802 | |

The results in Tab. 5 indicate that O-CoT consistently improves the answer accuracy for Object Counting questions. Though there remain errors after applying O-CoT, we will conduct refinement to fix them. Examples of Object Counting questions are provided in Appendix A.3.

## A.3  REFINEMENT DETAILS

We conduct refinement by passing raw LLM-generated responses into several human-defined filtering procedures based on the 3D scene graph. The refinement considers five raw response categories:

- Object Counting. The question concerns counting the target object.
- Object Existence.  The response claims the existence of objects, which can be actually either existent or non-existent.
- Object Non-existence. The response claims the non-existence of objects, which can be actually either existent or non-existent.
- Negative Response. The scene graph cannot provide a solid response to the question, which means the question cannot be answered and will be discarded.
- Response with ID. The response contains unexpected object IDs.

Specifically, we employ regular expression matching to detect errors in these five categories. And we also employ this method to correct the responses except for Response with ID, which will be rewritten by ChatGPT instead. The QA pair will be eliminated if multiple rounds of rewriting fail to remove the IDs. Tab. 6 and Tab. 7 show some examples of the responses subject to the above five categories as well as the effect of our refinement.

## A.4  STATISTICS OF RAW RESPONSES

Based on the aforementioned five raw response categories, we assess their quality by statistics and clarify the refinement effect accordingly.  In Tab. 8, we quantify the answer accuracy for Object Counting, Object Existence, and Object Non-existence in dialogue and QA tasks. Results of the two tasks are averaged over all 3DSSG scenes across 6 prompt seeds and 3 prompt seeds, respectively. For these three categories of responses, we can fix almost all the detected errors by referring to the scene graph. In Tab. 9, we present the proportion of Negative Response and Response with ID in the total set of responses. Negative responses will be removed, as well as the responses with remaining IDs after multiple rounds of rewriting. All results are based on the O-CoT method.

## A.5  SUBGRAPH SAMPLING

To enhance the diversity of the 3D scene graphs used for prompting, we perform subgraph sampling on the 3DSSG according to a sampling rate, which denotes the ratio of preserved nodes. The sampled subgraphs are used for generating scene captions and planning data. We analyze the distribution of node numbers across the 3DSSG dataset in Fig. 10 and set different sampling rates for scenes with

Table 6: **Examples of dialogue refinement**.

| Types | Raw Responses | Refined Responses |
|---|---|---|
| **Object Counting** | There are 3 chairs in the room. I see there are two washing machines in the bathroom. | There are 4 chairs in the room. I see there are 4 washing machines in the bathroom. |
| **Object Existence** | Yes, there is a cutting board in the kitchen. Yes, there is a computer and a monitor on the desk. However, the monitor is currently off. | No, there is no cutting board in the room. No, there is no computer in the room. |
| **Object Non-existence** | No, there is no stereo equipment in the room. I'm sorry, but I couldn't find a hair dryer in the bathroom. | Yes, there is a stereo equipment in the room. Yes, I found a hair dryer in the room. |
| **Negative Response** | No, there is nothing else mentioned in the scene graph. I'm sorry, but there is no mention of a mirror in the scene graph for the bathroom. | *The negative responses will be removed.* |
| **Response with ID** | You can place your backpack on the floor, to the left of the dining table-33. As for your bag, you can place it on the floor, to the left of the bed-10. | You can place your backpack on the floor, to the left of the dining table. As for your bag, you can place it on the floor, to the left of the bed. |

Table 7: **Examples of QA refinement**.

| Types | Raw Responses | Refined Responses |
|---|---|---|
| **Object Counting** | Q: How many chairs are in the room? A: 3 | Q: How many chairs are in the room? A: four |
| **Object Existence** | Q: Is there a mirror in the room? A: yes | Q: Is there a mirror in the room? A: no |
| **Object Non-existence** | Q: Is there an ironing board in the room? A: no | Q: Is there an ironing board in the room? A: yes |
| **Negative Response** | Q: What is the material of the bathtub? A: unknown Q: Where is the shampoo dispenser? A: unknown | *The negative responses will be removed.* |
| **Response with ID** | Q: Where is the mirror located? A: attached to wall-3, behind heater-18, to the left of shelf-19 | Q: Where is the mirror located? A: attached to a wall, behind a heater, to the left of a shelf |

Table 8: The accuracy of three types of raw responses from ChatGPT.

| Tasks | Object Counting | Object Existence | Object Non-existence |
|---|---|---|---|
| Dialogue | 0.7824 | 0.9314 | 0.2986 |
| QA | 0.7691 | 0.8999 | 0.2848 |

Table 9: The proportion of Negative Response and Response with ID in the total set of responses.

| Tasks | Negative Response | Response with ID |
|---|---|---|
| Dialogue | 0.0080 | 0.0360 |
| QA | 0.1045 | 0.0221 |

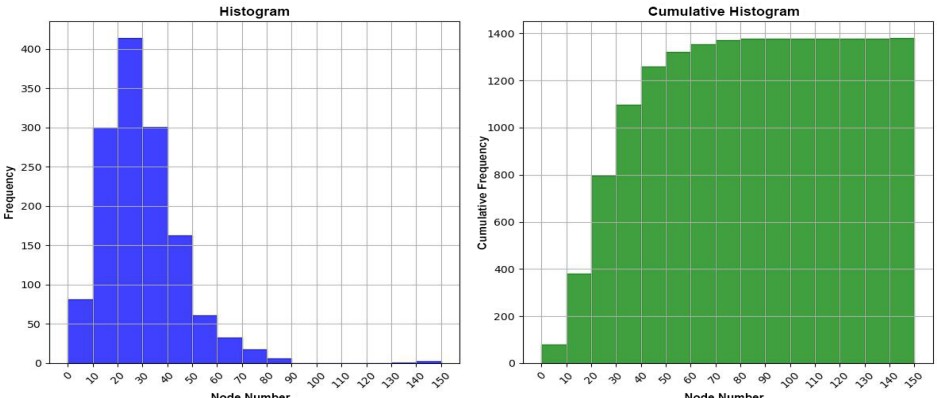

Figure 10: **The distribution of node numbers for 3DSSG scenes.** The node number represents the number of objects in a scene.

Table 10: **Sampling rates for scenes with different node numbers.** The hyphen denotes a sweep of sampling rates, *e.g.*, "0.7-0.9" means "0.7,0.8,0.9".

| Node Number | 10-20 | 20-30 | 30-40 | 40-50 | 50-60 | 60-70 | >70 |
|---|---|---|---|---|---|---|---|
| Sampling Rate | 0.8,0.9 | 0.7-0.9 | 0.6-0.9 | 0.6-0.9 | 0.5-0.9 | 0.5-0.9 | 0.4-0.9 |

different numbers of nodes in Tab. 10. For each sampling rate, we set 4 random prompt seeds to further enhance the diversity of prompted data.

To verify whether the subgraph sampling strategy can maintain the consistency and diversity of scene captions, we generate scene captions for the same scene using both the full graph and subgraph. We then employ GPT-4 (OpenAI, 2023) to evaluate the similarities and differences between the two captions. The results in Tab. 11 indicate that our subgraph sampling strategy can maintain both consistency and diversity.

### A.6 SCENE-GRAPH-BASED PROMPTING *vs.* BOX-BASED PROMPTING

In this section, we provide a comparative analysis of scene-graph-based prompting and box-based prompting (Hong et al., 2023). We refer the readers to Figure 6 in 3D-LLM (Hong et al., 2023) for details of the box-based prompting method. Fig. 11 shows the contents of two methods. To present a fair comparison between the two methods, we prompt with 1) demonstrations that have similar content under the same scene (see Fig. 12) and 2) identical new scene queries. Since 3D-LLM does not elaborate on attribute-related prompts, we mainly compare the spatial relations in the responses. As shown in Fig. 13, we highlight some spatial relations in red. The comparison shows that our method provides more diverse and reliable spatial relations, which are important for 3D scene understanding.

### A.7 DATASET STATISTICS

We provide statistics on the instruction-tuning datasets. We visualize the distribution of the question types in 3RQA (Fig. 14) and 3RDialog (Fig. 15). The pie chart's inner circle represents the first word of the questions, while the outer circle accounts for the second or third word in the corresponding questions. The results show that the questions cover the attributes and spatial relations of the objects, as well as high-level topics such as room types and functionalities.

We also provide statistics of the root noun-verb pairs for instructions and responses in 3RDialog and 3RPlan, as shown in Fig. 16–19.

## B ACTION TOKENIZATION

To empower LEO to exert control over an embodiment or a robot, we encode all actions within the context of Object Navigation (Ramrakhya et al., 2022) and CLIPort (Shridhar et al., 2021) tasks

Table 11: **The comparative analysis of full graph prompting and subgraph prompting.**

| Settings | Generated Scene Captions |
|---|---|
| **Full Graph** | In this scene, there is a colorful and flat white wall. A small TV stand is positioned in front of a low upholstered couch, to the left of a brown lamp, and to the right of a wooden table. The couch is covered with a pink padded square pillow and a patterned blanket. A light hangs on the wall and another light is supported by the table. There is a rectangular glass window attached to the wall, behind which is a yellow curtain. The room has a comfortable and cozy atmosphere, suitable for relaxation and entertainment. |
| **Sampling Rate 0.6** | In this scene, there is a colorful white wall attached to a brown wooden floor. The couch is upholstered and low in size, located to the right of a small TV stand and a table. A red blanket covers the couch, and there are patterned, padded square pillows on the couch. A black glass window is attached to the wall, and there are artificial lamps standing on the floor and close to the couch. The lamps are also near a table and to the right of the TV stand. The scene suggests a comfortable living room with a cozy couch, pillows, and a blanket, suitable for relaxation and entertainment. |
| **GPT-4 Evalutaion** | **Similarities:** 1. Both scenes describe a living room setting, emphasizing comfort and coziness suitable for relaxation and entertainment. 2. Both rooms have a white wall. 3. Each scene features a couch and a TV stand. 4. Both scenes describe a blanket and a padded square pillow on the couch. 
 **Differences:** 1. The first scene has a brown wooden floor, while the floor is not mentioned in the second scene. 2. The first scene has a red blanket on the couch; the second has a patterned blanket but doesn't specify the color. 3. The first scene describes the lamps as "standing on the floor", while the second mentions one light hanging on the wall and another supported by the table. 4. The second scene includes a yellow curtain behind the window, which the first scene does not mention. 
 **Summary:** Overall, both summaries provide a similar thematic view of a comfortable living room but differ in the specific arrangement and color details of the items within the room. |

---

**Box-based Content**

wall:[-0.66, 0.853, -0.329], floor:[0.291, 0.454, -1.533], ceiling:[0.3, 0.955, 0.9], wall:[0.997, 0.577, -0.353], light:[0.213, 0.59, 0.905], wall:[0.971, 3.168, -0.351], window:[0.943, 3.385, 0.074], board:[-0.649, -0.117, -1.183], desk:[0.696, 2.259, -0.987], box:[-0.395, 0.64, -1.33], bowl:[0.631, 3.071, -0.803], box:[0.797, 3.121, -0.91]

---

**Scene-Graph-based Content**

{'wall-1': {'relations': ['attached to floor-2'], 'attribute': {'shape': 'flat', 'lexical': 'architectural', 'color': 'white'}}, 'floor-2': {'relations': [], 'attribute': {'material': 'plastic', 'shape': 'flat', 'lexical': 'inside', 'color': 'blue'}}, 'ceiling-3': {'relations': ['attached to wall-1', 'attached to wall-4', 'attached to wall-7'], 'attribute': {'shape': 'flat', 'lexical': 'overhead', 'color': 'white'}}, 'wall-4': {'relations': ['attached to floor-2'], 'attribute': {'shape': 'flat', 'lexical': 'architectural', 'color': 'white'}}, 'light-6': {'relations': ['hanging on ceiling-3'], 'attribute': {'state': 'off'}}, 'wall-7': {'relations': ['attached to floor-2'], 'attribute': {'shape': 'flat', 'lexical': 'architectural', 'color': 'white'}}, 'window-8': {'relations': ['attached to wall-7', 'behind desk-10'], 'attribute': {'material': 'glass', 'color': 'dark', 'shape': 'rectangular', 'state': 'closed'}}, 'board-9': {'relations': ['lying on floor-2', 'to the left of desk-10', 'close by box-11'], 'attribute': {'shape': 'flat', 'lexical': 'flat', 'color': 'brown'}}, 'desk-10': {'relations': ['standing on floor-2', 'in front of window-8', 'to the right of board-9', 'to the right of box-11', 'close by box-11'], 'attribute': {'other': 'rigid', 'size': 'narrow'}}, 'box-11': {'relations': ['standing on floor-2', 'close by board-9', 'close by desk-10', 'to the left of desk-10', 'in front of box-15', 'to the left of box-15'], 'attribute': {'state': 'written on', 'shape': 'rectangular', 'lexical': 'rectangular', 'other': 'rigid', 'size': 'tall'}}, 'bowl-14': {'relations': [], 'attribute': {}}, 'box-15': {'relations': ['standing on desk-10', 'to the right of box-11', 'behind box-11'], 'attribute': {'color': 'dark', 'shape': 'rectangular', 'lexical': 'rectangular', 'size': 'small'}}}

Figure 11: Comparison of the content between box-based and scene-graph-based prompting.

using the least frequently employed language tokens. Specifically, for the Object Navigation task, we allocate 4 tokens to represent actions of *move forward*, *turn right*, *turn left*, and *stop*. For the CLIPort task, we use a total of 516 tokens to discretize action poses, with 320 tokens dedicated to the x-axis pose bins, 160 tokens for the y-axis pose bins, and 36 tokens for the z-rotation bins.

## C  DATA EXAMPLES

Please refer to Tabs.  for examples of our dataset.

**Box-based Prompting**

**Demonstration #1:**
Human: Do you think the room is suitable for me to read a book?
Robot: Let me check the arrangement of the room. There is a light in the room, which can provide adequate lighting. There is a desk for you to put your book. However, there is no chair of sofa in the room. It means you cannot seat down and rest in the room.
Human: Thanks a lot. Can you describe the position of the desk?
Robot: Sure! The desk is in front of the window.
Human: Thanks a lot.
Robot: You are welcome!

**Demonstration #2:**
Human: Can you help me print a document?
Robot: Sure! There is a printer in the room. You can use it for printing.
Human: Ok. I also need a laptop to download the document.
Robot: Ok. I find a laptop on the table. Do you need any other help?
Human: Thanks a lot.
Robot: You are welcome!

**Scene-Graph-based Prompting**

**Demonstration #1:**
Dialogue Context: high level task: verify if the furniture arrangement is suitable for reading. low level task: check specific object.
Human: Do you think the room is suitable for me to read a book?
Thought:light-6,desk-10
Robot: Let me check the arrangement of the room. There is a light in the room, which can provide adequate lighting. There is a desk for you to put your book. However, there is no chair of sofa in the room. It means you cannot seat down and rest in the room.
Human: Thanks a lot. Can you describe the position of the desk?
Thought: desk-10
Robot: Sure! The desk is in front of the window.
Human: Thanks a lot.
Thought: no specific object
Robot: You are welcome!

**Demonstration #2:**
Dialogue Context: high level tasks: find proper objects to print a document.
Human: Can you help me print a document?
Thought: printer-30
Robot: Sure! There is a printer in the room. You can use it for printing.
Human: Ok. I also need a laptop to download the document.
Thought: laptop-28
Robot: Ok. I find a laptop on the table. Do you need any other help?
Human: Thanks a lot.
Thought: no specific object
Robot: You are welcome!

Figure 12: The demonstrations for prompting, which are designed to be similar for a fair comparison.

**Box-based Prompting**

**Response:**
Human: Can you help me find a towel?
Robot: Of course! I found three towels in the room. One is near the bathtub, one is near the sink, and another one is near the stool. Which one would you like to use?
Human: The one near the sink, please.
Robot: Alright. I will fetch the towel near the sink for you. Is there anything else I can assist you with?
Human: No, thank you. That's all.
Robot: You're welcome! If you need any further assistance, feel free to ask.

**Scene-Graph-based Prompting**

**Response:**
Dialogue Context: high level task: describe the objects in the bathroom.
Human: Can you describe the objects in the bathroom?
Thought: bathtub-8, sink-12, toilet-15, mirror-4
Robot: Sure! In the bathroom, there is a bathtub made of ceramic, with a shiny texture and a white color. There is also a sink with an oval shape and a shiny texture. The toilet is white and seat down. There is a mirror attached to the wall, which is polished. Is there anything specific you want to know about these objects?
Human: Is there a towel in the bathroom?
Thought: towel-10
Robot: Yes, there is a folded together rectangular towel in the bathroom. It is located to the right of the door.

Figure 13: The responses of two prompting methods. Descriptions highlighted in red show our method leads to more flexible and reliable spatial relations.

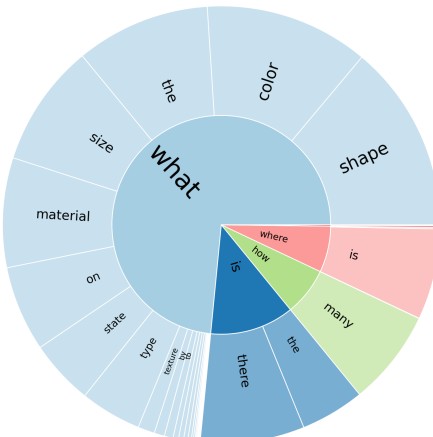

Figure 14: Question types: 3RQA.

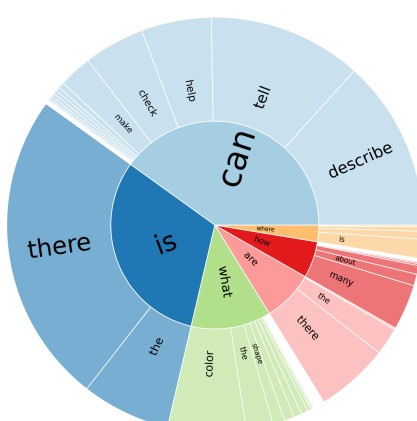

Figure 15: Question types: 3RDialog.

# D MODEL DETAILS

## D.1 PROMPTS

The first portion of prompts sent into the LLM is a **system message**. It consists of two parts: a role prompt and a situation prompt. The role prompt is the same for all tasks:

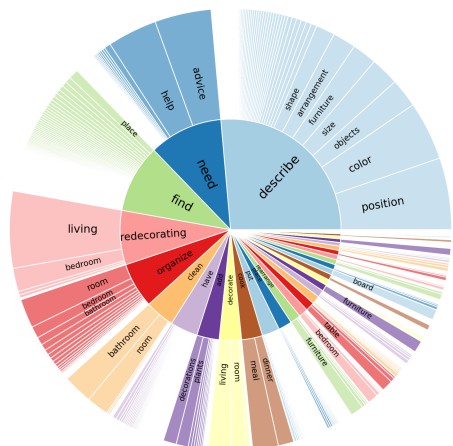

Figure 16: Noun-verb pairs: 3RDialog instruction.

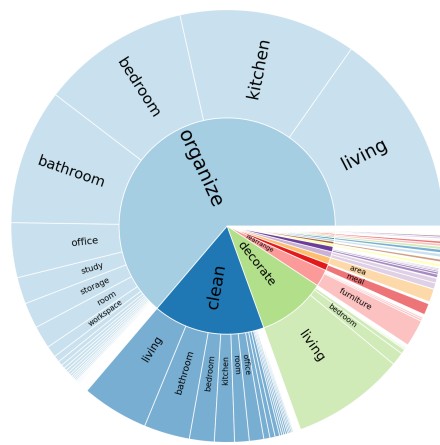

Figure 17: Noun-verb pairs: 3RPlan instruction.

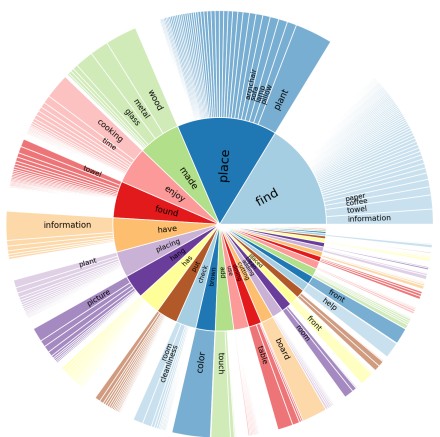

Figure 18: Noun-verb pairs: 3RDialog response.

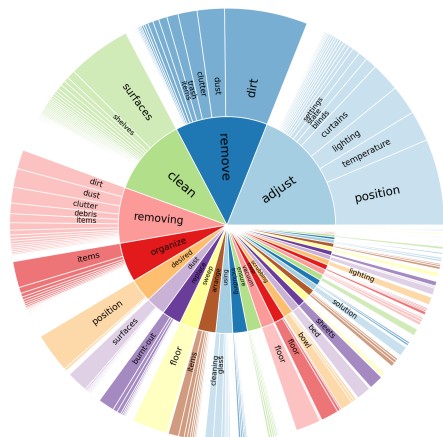

Figure 19: Noun-verb pairs: 3RPlan response.

> You are an AI visual assistant situated in a 3D scene. You can perceive (1) an ego-view image (accessible when necessary) and (2) the objects (including yourself) in the scene (always accessible). You should properly respond to the USER's instructions according to the given visual information.

The situation prompt begins with a common sentence:

> You are at a selected location in the 3D scene.

For SQA3D (Ma et al., 2023), the situation prompt is further extended with the situation description in the dataset. The situation prompt is only used jointly with the embodiment token to support tasks that require information about the embodiment. Details can be found in Appendix D.2.1.

Next are the **visual tokens**, including **2D image tokens** and **object-centric 3D tokens**. Each token sequence is interleaved within text tokens and starts with a text prefix.

> Ego-view image: {IMAGE_TOKENS}
> Objects (including you) in the scene: {OBJECT_TOKENS}

The last portion of prompts is a **task-specific instruction**. For **object-level caption** and **object-in-the-scene caption**, we randomly chose one sentence from 151 sentences to be the instruction.

Some examples can be found in Tab. 12. For **scene-level caption**, we randomly choose one from 183 instructions. Examples can be found in Tab. 13. For **3D question answering** task, we simply use the question as the instruction. The dialog history is used as the instruction for **3D dialogue** to provide continuity across multiple rounds of interactions. A planning instruction pool consisting of 202 instructions is introduced for **scene-aware task planning** and we randomly choose one from it as done in the caption tasks. Examples from the pool can be found in Tab. 14. The chosen instruction is further followed by an instruction that specifies the task, *e.g.*, *set up a home office*.

With past action tokens {PAST_ACTIONS} appended at the end, the instruction for **embodied navigation** is as follows, where {GOAL} stands for the goal specified by the target object name:

> The task is navigation. Your goal is to find {GOAL} by moving around in the scene. Past actions: {PAST_ACTIONS}.

The instruction for **robotic manipulation** is similar to the one in **embodied navigation**. Here {GOAL} is the task description in CLIPort:

> The task is manipulation. Your goal is to {GOAL}. Past actions: {PAST_ACTIONS}.

Table 12: **Examples from our object-level caption instruction set.**

> "Produce a description for the object at the chosen spot in the 3D scene.",
> "How would you depict the object located at the selected point in the 3D environment?",
> "Formulate a description of the item at the picked position within the 3D scene.",
> "How would you describe the entity at the designated location in the 3D backdrop?",
> "Can you detail the object situated at the selected point in the 3D setting?",
> "Compose a narrative for the object at the chosen locale within the 3D environment.",
> "What does the object at the specified position in the 3D visualization look like?",
> "Provide a description for the item located at the marked site in the 3D world.",
> "How would you illustrate the object placed at the selected spot in the 3D landscape?",
> "Craft a depiction of the object at the pinpointed location within the 3D territory.",
> "What kind of object is illustrated at the identified site in the 3D tableau?",
> "Develop a description of the object at the specified position in the 3D backdrop.",
> "What is the entity's detail at the highlighted site in the 3D view?",
> "Write up a description of the entity at the selected spot in the 3D realm.",
> "What does the object look like at the pinpointed location in the 3D space?",
> "Detail the entity located at the chosen position within the 3D scene.",
> "Can you explain the essence of the object at the selected spot in the 3D zone?",

### D.2 FEATURE ENCODING

We have several modules to encode the multi-modal features.

- **Object-centric 3D token embedding.** The encoder for 3D object-centric point clouds is a PointNet++ (Qi et al., 2017) pre-trained on ScanNet (Dai et al., 2017) with object-classfication task. We sample 1024 points for every object as in Chen et al. (2022). The architecture parameters all remain the same with Chen et al. (2022). We freeze the PointNet++ for empirically better results.
- **Spatial Transformer (Chen et al., 2022).** Spatial Transformer is a modified transformer architecture that explicitly encodes spatial relations between object pairs. Specifically, consider the vanilla self-attention (Vaswani et al., 2017) mechanism which takes as input a feature matrix $X \in \mathbf{R}^{N \times d}$, where $N$ stands for the number of tokens and $d$ is the feature dimension. Vanilla self-attention first compute $Q = XW_Q, K = XW_K, V = XW_V$ from $X$ using learnable projection matrices $W_Q, W_K, W_V \in \mathbf{R}^{d \times d_h}$ where $d_h$ stands for the output feature dimension. Then the attention weight matrix is computed by $(\omega_{ij}^o)_{N \times N} = \Omega^o = softmax(\frac{QK^T}{\sqrt{d_h}})$ and finally used for re-weighting $\Omega^o V$. The intuition of Spatial Transformer is that we can re-scale the elements $\omega_{ij}^o$ in the weight matrix $\Omega^o$.

Table 13: **Examples from our scene-level caption instruction set.**

> "Describe this scene.",
> "Generate a description of this scene.",
> "Generate a caption of this scene.",
> "Can you describe the scene?",
> "Can you generate a description of the scene?",
> "Can you generate a caption of the scene?",
> "Summarize this scene.",
> "Provide an outline of this 3D scene's characteristics.",
> "How would you describe the 3D scene?",
> "How would you summarize this scene?",
> "Convey a summary of the 3D structure of this scene.",
> "How would you interpret this 3D scene?",
> "Offer a summary of the 3D scene.",
> "Can you describe this scene in detail?",
> "I'm interested in this scene, can you explain?",
> "What is this scene made of?",
> "Could you provide more info about this scene?",

Table 14: **Examples from our planning instruction pool.**

> "Plan for the task",
> "Can you come up with a plan for this task",
> "How can we do this task, provide a step-by-step plan",
> "Draft a plan for completing this task",
> "Detail a strategy for the task",
> "What's the best plan for this task",
> "Draw out a procedure for the task",
> "Lay out the steps for this task",
> "Could you devise a plan for the task",
> "Show me a plan for this task",
> "I need a plan for the task",
> "Sketch a plan for the task at hand",
> "Set up a plan for this",
> "Recommend a plan for this task",
> "Offer a strategy for this task",
> "Design a blueprint for the task",
> "Outline the approach for this task",

In the object-centric reasoning setting, the input feature matrix is $O \in \mathbf{R}^{N \times d}$. Consider an object pair $(O_i, O_j)$ with their geometric centers $c_i, c_j$. Spatial Transformer (Chen et al., 2022) computes the Euclidean distance $d_{ij} = ||c_i - c_j||_2$ and the horizontal and vertical angles $\theta_h, \theta_v$ of the line connecting $c_i$ and $c_j$. The spatial feature between the two objects $(O_i, O_j)$ is a 5-dimensional vector $f_{ij} = [d_{ij}, \sin(\theta_h), \cos(\theta_h), \sin(\theta_v), \cos(\theta_v)]$. To combine this feature with objects, the spatial attention computes $\omega_{ij}^s = g_i f_{ij}$ where $g_i = W_S^T o_i$ is a 5-dimensional vector. The spatial attention further reweights the original self-attention weight matrix as

$$\omega_{ij} = \frac{\sigma(\omega_{ij}^s) exp(\omega_{ij}^o)}{\sum_{l=1}^{N} \sigma(\omega_{il}^s) exp(\omega_{il}^o)}.$$

Readers are referred to Chen et al. (2022) for more details. In summary, Spatial Transformer explicitly computes pairwise spatial relations and fuses them with vanilla self-attention to provide better spatial reasoning ability. We use a three-layer Spatial Transformer with 8 heads to process

the object-centric features produced by PointNet++ and output object tokens for LLM. For other settings, We follow all the default hyperparameters in Chen et al. (2022).

- **2D token embedding.** We use OpenCLIP ConvNext-base model (Liu et al., 2022) pre-trained on LAION2B (Schuhmann et al., 2022) to process the egocentric 2D image.

- **CLIP fusion.** To enhance the alignment between visual tokens and instruction tokens, we use the text encoder from CLIP (Radford et al., 2021) to process the instruction tokens to obtain a global feature of the instruction. Next, we update the visual tokens with the element-wise product between the CLIP instruction feature and each image & object token embedding.

### D.2.1 Embodiment Encoding

In addition to the egocentric 2D input, we introduce an embodiment token to help LEO reason in an embodiment-aware fashion. We find it useful to use it together with the situation prompt and 2D egocentric input. Specifically, an embodiment token $e$ is introduced in **embodied navigation**, **embodied reasoning**, and **object-in-the-scene caption** tasks. Specifically, $e$ is a learnable embedding that will be inserted into the 3D object list.

So what does embodiment information mean in these tasks? In **embodied navigation**, it means the agent's position and orientation in the scene, which can be derived from a GPS and a compass sensor. The orientation of the agent is further represented by a rotation which is Fourier-embedded and mapped to a feature vector $r$ by a linear layer. It is the same in **embodied reasoning** task. In the **object-in-the-scene caption** task, we assume the agent is situated at the location of the object that is being referred to. Therefore, embodiment information also means the location of the referred object. We obtain this location by randomly choosing a spot inside the referred object bounding box. To sum up, we could simply treat the embodiment token as a special *self object*, where its object embedding is learnable, and its location/orientation corresponds to the actual or assumed "agent".

After inserting the embodiment token, we obtain a new 3D object token list: $e, s_{3D}^{(1)}, s_{3D}^{(2)}, \ldots, s_{3D}^{(N)}$, where $s_{3D}^{(i)}, i \in \{1, 2, \ldots, N\}$ are 3D object token embeddings produced by PointNet++, along with location specified for each object (including the *self-object*). We can concatenate them together to get a feature matrix $O \in \mathbf{R}^{(N+1) \times d}$ and send them to the Spatial Transformer to explicitly fuse the spatial information of all the 3D objects and the self-object.

### D.3 LLM Hyperparameters

We set the maximum output length of our Vicuna-7B to be 256. The maximum context length is also set to 256 and if the length of the input is greater than 256, we truncate it to 256 by deleting tokens from the left (*i.e.*, only the rightmost 256 tokens are preserved). We set rank and $\alpha$ in LoRA (Hu et al., 2022) to be 16 and the dropout rate to be 0. LoRA is implemented for all the projection matrices in the LLM, *i.e.*, $(W_q, W_k, W_v, W_o)$ in attention modules and $(W_{gate}, W_{up}, W_{down})$ in MLPs.

The hyperparameters for beam search during inference are as follows:

Table 15: Hyperparameters for LEO inference.

| Beam search hyperparameter | Value |
| --- | --- |
| Number of beams | 5 |
| maximum output length | 256 |
| minimum output length | 1 |
| top p | 0.9 |
| repetition penalty | 3.0 |
| length penalty | 1 |
| temperature | 1 |

## E  Alignment Setup

The hyperparameters for the first-stage 3D vision-language alignment are presented in Tab. 16

Table 16: Hyperparameters for the alignment stage.

| Hyperparameter | Value |
|---|---|
| Optimizer | AdamW |
| Weight Decay | 0.05 |
| betas | [0.9, 0.999] |
| Learning Rate | $3 \times 10^{-4}$ |
| Warmup Steps | 400 |
| Number of Workers | 4 |
| Parallel Strategy | DDP |
| Type of GPUs | NVIDIA A100 |
| Number of GPUs | 4 |
| Accumulate Gradient Batches | 5 |
| Batch Size/GPU (total) | 4 (80) |
| Training Precision | bfloat16 |
| gradient norm | 5.0 |
| epochs | 10 |

## F    INSTRUCTION-TUNING SETUP

The hyperparameters for 3D VLA instruction tuning are presented in Tab. 17

Table 17: Hyperparameters for the instruction-tuning stage.

| Hyperparameter | Value |
|---|---|
| Optimizer | AdamW |
| Weight Decay | 0.05 |
| betas | [0.9, 0.999] |
| Learning Rate | $3 \times 10^{-5}$ |
| Warmup Steps | 400 |
| Number of Workers | 4 |
| Parallel Strategy | DDP |
| Type of GPUs | NVIDIA A100 |
| Number of GPUs | 4 |
| Accumulate Gradient Batches | 5 |
| Batch Size/GPU (total) | 4 (80) |
| Training Precision | bfloat16 |
| gradient norm | 5.0 |
| epochs | 10 |

## G    ABLATION DETAILS

**Data ablation.**    We present the numerical results in Tab. 18 as complements to Fig. 4(a).

**Model ablation.**    We also make some explorations in model ablation and simply present qualitative findings here. For the point cloud encoder, we choose Point-BERT (Yu et al., 2022) as an alternative to the default PointNet++ (Qi et al., 2017). We utilize the checkpoint from PointLLM (Xu et al., 2023), which has adapted Point-BERT to 6-channel (XYZRGB) input and learned a language-aligned representation for 3D objects. Despite larger capacity, Point-BERT shows significantly worse performances than PointNet++ as the point cloud encoder. Similarly, for the Spatial Transformer and LLM, we ablate with different model scales but find no improvement. The influence of different modules remains an interesting question that deserves further exploration.

## H    EVALUATION DETAILS

### H.1    3D QUESTION ANSWERING

**Rationality of QA evaluation protocol.**    We argue that exact match (EM), as a conventional metric for 3D QA, is unsuitable for evaluating the open-ended answer generated by LLMs. For example,

Table 18: Evaluation results of LEO with different data configurations for training. We test models on the validation set of Scan2Cap and ScanQA, and the held-out test sets of other datasets. We report exact match scores for QA tasks and SentenceSim for all other tasks. * indicates datasets that belong to LEO-align. Figures in gray indicate zero-shot transfer.

| | Cap3D* | SceneCap* | Scan2Cap | ScanQA | SQA3D | 3RQA | 3RPlan | 3RDialog |
|---|---|---|---|---|---|---|---|---|
| *NoAlign* | 20.5 | 23.6 | 52.9 | 35.3 | 50.9 | 46.8 | 69.2 | 76.4 |
| *PartialData* | 23.0 | 61.5 | 62.2 | 30.2 | 45.4 | 47.3 | 69.3 | 76.4 |
| *ScanNetOnly* | 34.5 | **67.0** | **65.2** | **37.8** | **54.4** | 32.5 | 33.2 | 63.2 |
| *NoAct (VL)* | **35.9** | 65.5 | 64.9 | 36.8 | 54.1 | **61.3** | **74.1** | **79.6** |
| *Full (VLA)* | 12.9 | 64.7 | 62.1 | 37.0 | 51.5 | 55.9 | 73.5 | 78.4 |

Table 19: Examples from ScanQA validation set, showing the rationality of our refined exact match protocol.

| Question | Ground-truth answer | Generated answer | Strict EM | Refined EM |
|---|---|---|---|---|
| What color is the chair in the kitchen? | dark brown | brown | ✗ | ✓(case 2) |
| What is under the long kitchen counter? | kitchen cabinets | brown rectangular kitchen cabinets | ✗ | ✓(case 2) |
| What type of refrigerator is on the right of a kitchen counter? | stainless steel refrigerator | stainless steel | ✗ | ✓(case 2) |
| Where is the beige wooden desk placed? | up against wall | against wall | ✗ | ✓(case 2) |
| What color does the sofa look? | it looks black | black | ✗ | ✓(case 2) |
| Where is the black office chair located? | in front of desks | in front of desk | ✗ | ✓(case 2) |
| What is in the corner by windows? | book shelf | bookshelf | ✗ | ✓(case 2) |
| Where is the chair pulled into? | table | under table | ✗ | ✓(case 3) |
| How many chairs are to the left of the table? | 4 | 4 chairs | ✗ | ✓(case 3) |
| What objects are sitting on the black couch? | pillow | pillows | ✗ | ✓(case 3) |
| Where are the two different size tables located in room? | in center | in center of room | ✗ | ✓(case 3) |
| Where is the laptop located? | desk | on desk | ✗ | ✓(case 3) |
| Where is the soap dispenser mounted | above sink | on wall above sink | ✗ | ✓(case 3) |

given the question "*On what side of the towel is a bathroom curtain?*" with ground-truth answer "*left side of towel*", it is never wrong to answer "left". However, this will be deemed incorrect if we adopt the strict exact match protocol. Such a misjudgment is quite likely to occur when evaluating the answers from LLMs. By contrast, the classifier heads for QA (*e.g.*, MCAN) are less affected because they collect all possible answers in advance to formulate the QA as a close-set classification problem. Hence, we refine the strict exact match protocol as follows.

```
"""
code for QA protocols
pred: str
gts: List[str]
"""

def strict_em(pred, gts):
    for gt in gts:
        if pred == gt:
            # case 1
            return True

def refined_em(pred, gts):
    for gt in gts:
        if pred == gt:
            # case 1
            return True
        elif ''.join(pred.split()) in ''.join(gt.split()):
            # case 2
            return True
        elif ''.join(gt.split()) in ''.join(pred.split()):
            # case 3
            return True
    return False
```

In a nutshell, we squeeze the `pred` and `gt`, and then check whether one is a subset of the other. To justify our refined exact match protocol, in Tab. 19 we provide some representative examples in the ScanQA validation set. Despite the improvements, we speculate such a simple refinement is still insufficient for a sound evaluation metric considering the flexibility of human language.

## H.2 Embodied Navigation

To construct our training set, we adopt all 57 scenes in the MP3D `ObjNav` training split (Savva et al., 2019; Ramrakhya et al., 2022) and generate ~60K shortest-path navigation episodes. The evaluation is conducted on the original validation split of the MP3D `ObjNav` task and the newly introduced HM3D `ObjNav` task (Ramakrishnan et al., 2021).

In contrast to most `ObjNav` agents that utilize recurrence through either RNN (Ramrakhya et al., 2022) or DT-style Transformer (Suglia et al., 2021), LEO only employs a simplistic feed-forward policy, *i.e.*, the Transformer in LEO only takes in the instruction, current state (2D and 3D observation), and past 4 actions, and predicts the next action, similar to RT-2 (Brohan et al., 2023). Therefore, the only information relayed from the past is about past actions. The absence of recurrence in LEO's acting policy is indeed the result of a trade-off between better performances and training efficiency. We will commit to exploring the possibility of looping in more sophisticated policy architectures (*e.g.*, recurrence) in future work.

# I Additional Results

## I.1 Embodied Acting

**Quantitative results of `ObjNav`.** We provide additional results of LEO 1) generalizing to unseen objects on MP3D, and 2) learning with 70K human demonstrations provided by Habitat-web (Ramrakhya et al., 2022) instead of shortest path. Below is a list of the objects used during training (seen) and for OOD evaluation (unseen). Evaluation results are shown in Tab. 20. Note that the baseline Habitat-web is unable to generalize to novel objects as it uses categorical embedding rather than natural language to represent object goals.

```
# Objects (seen)
"gym_equipment", "tv_monitor", "picture", "counter",
"chair", "cabinet", "table", "stool", "plant", "towel",
"sofa", "cushion", "sink", "fireplace", "toilet", "seating",
"chest_of_drawers", "bed", "shower", "bathtub", "clothes"

# Objects (unseen)
"shelf", "pillow", "lamp", "box", "desk", "refrigerator",
"vase", "armchair"
```

Table 20: **Results on object navigation with OOD objects and human demonstrations.** Note that the baseline Habitat-web is unable to generalize to MP3D-unseen as it uses categorical embedding rather than natural language to represent object goals.

|  | MP3D-seen | | MP3D-unseen | |
| --- | --- | --- | --- | --- |
|  | Success(↑) | SPL(↑) | Success(↑) | SPL(↑) |
| Habitat-web (shortest) | 4.4 | 2.2 | - | - |
| Habitat-web (70k demo) | **35.4** | 10.2 | - | - |
| LEO (shortest) | 23.1 | **15.2** | **11.1** | **9.6** |
| LEO (70k demo) | 7.1 | 5.3 | 8.9 | 8.6 |

The results show that LEO can generalize to novel objects. On the other hand, human demonstrations include more explorations, compared with shortest-path data. Therefore, it will be much harder for agents without a recurrent module (*e.g.*, LEO) to learn from human demonstrations (see Appendix H.2), leading to significantly weaker performances.

**Qualitative results**. We provide more qualitative results of robotic manipulation and embodied navigation in the supplementary video.

Table 21: **Additional qualitative results on Scan2Cap validation set**.

| Scene | Instruction | Response | Ground Truth |
|---|---|---|---|
|  | Develop a description of the object at the specified position in the 3D backdrop. | This is a black keyboard. It is in front of two monitors. | The keyboard sits atop the workstation desk. It is located in front of the left-side monitor. |
|  | Focus on the object at the given 3D location and describe it. | This is a white pillow. It is on the couch. | It is a small pillow located on the couch. You can notice it directly on your left when walking through the door into the room. |
|  | Provide a clearer description of the object at that specific 3D spot. | This is a white cabinet. It is to the right of a couch. | This is a white ottoman. It is located to the right of the black couch. |
|  | What does the object at the specified position in the 3D visualization look like? | There is a rectangular brown door. It is next to a couch. | There is a door in the middle of the northern wall next to a metal cabinet. The door is shaped like a brown rectangle and as a dark yellow colored door-knob. |

## I.2 SCAN2CAP

We provide additional qualitative results on Scan2Cap validation set in Tab. 21. The results show that LEO can correctly refer to the queried object and provide accurate descriptions, including spatial relationships with other objects. However, LEO's responses are confined to simple formats that lack diversity. How to unlock more flexible responses while maintaining accuracy can be a direction for future research.

## I.3 SCANQA

We provide additional qualitative results on ScanQA validation set in Tab. 22 and categorize the responses into several types:

- *Wrong*. The response is inaccurate and deemed wrong.
- *Wrong but reasonable*. The response is deemed wrong but is reasonable to some extent, probably due to ambiguities in the scene. Consider the second case in Tab. 22. There are many objects such as a coat rack, a coat, and a mini fridge-shaped cabinet on the right side of the organizer. Though LEO's response "mini fridge" does not match the ground truth "coat rack", it is consistent with the 3D scene layout.
- *Wrong but accurate*. The response is accurate according to the scene but is deemed wrong due to imperfect ground truth annotations.
- *Correct*. The response is accurate and deemed correct.

Table 22: **Additional qualitative results on ScanQA validation set**.

| Scene | Question | Response | Ground Truth | Response Type |
|---|---|---|---|---|
| | What is placed on the black chair out of the three chairs present in the room? | pillow | tan jacket, guitar box like object | *Wrong* |
| | What is next to the organizer shelf? | mini fridge | coat rack | *Wrong but reasonable* |
| | Where is the dresser located? | to right of door | underneath television next to black guitar case and green guitar case, under tv set next to guitar cases | *Wrong but accurate* |
| | What kind of desk is the electric piano on? | wooden desk | wooden desk | *Correct* |
| | How many monitor sits on top of the desk? | 2 | 2 | *Correct* |
| | What is on the left side of the brown desk? | black computer monitor | monitor / computer monitor | *Correct and more accurate* |

- *Correct and more accurate*. The response is more accurate than the ground truth annotations.

## I.4  SQA3D

We provide additional qualitative results on SQA3D test set in Tab. 23 and follow the aforementioned response types. The embodied reasoning in SQA3D requires the understanding of not only the scene but also the situation of embodiment. In Tab. 23, answering "What am I sitting at?" necessitates that LEO accurately identifies the objects at its current location. And the response to "How many beds are in front of me?" indicates that LEO can reason based on the understanding of its orientation.

Table 23: **Additional qualitative results on SQA3D test set**.

| Scene | Question | Response | Ground Truth | Response Type |
|---|---|---|---|---|
|  | What is the only orange object on my left side? | backpack | jacket | *Wrong* |
|  | Is there anything on the desk? | yes | laptop | *Wrong but accurate* |
|  | What type of bed is the one on the right? | loft bed | loft bed | *Correct* |
|  | What am I sitting at? | desk | desk | *Correct* |
|  | How many beds are in front of me? | one | one | *Correct* |

Table 24: **Examples from our datasets**. We provide the tasks, 2D images (optional), 3D point clouds, and instructions as the input of LEO, as well as the expected responses of LEO.

| Dataset & Task | 2D & 3D input | Instruction | Response |
|---|---|---|---|
| LEO-align object captioning |  | USER: Explain the object at the pinpointed location in the 3D setting. ASSISTANT: | a yellow metal cage with a fence around it. |
| LEO-align object referring |  | USER: Give me a snapshot of the object at the chosen 3D location. ASSISTANT: | find the office chair that is near the copier |
| LEO-align scene captioning |  | USER: Relay a brief summary of this scene's elements. ASSISTANT: | In this scene, there is a gray flat picture hanging on the wall, standing in front of a windowsill. There is also a rectangular white padded pillow lying on the bed, close to a big rectangular white padded pillow. The room has a white architectural wall and a flat overhead ceiling. The bed is tidy and white, located in front of a narrow tall commode. The curtains are hanging on the wall. The room has a calm and comfortable atmosphere, suitable for relaxation and sleep. |

Table 25: **Examples from our datasets**. Continued.

| | | | |
|---|---|---|---|
| LEO-instruct 3D captioning | 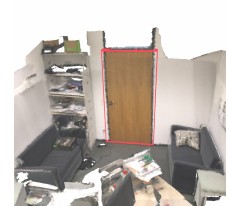 | USER: What does the object at the specified position in the 3D visualization look like? ASSISTANT: ASSISTANT: | there is a door in the middle of the northern wall next to a metal cabinet. the door is shaped like a brown rectangle and as a dark yellow colored doorknob. |
| LEO-instruct 3D QA | 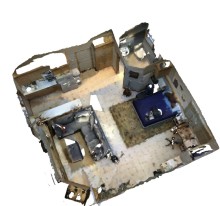 | USER: What is in the right corner of room by curtains? ASSISTANT: | brown cabinet with tv sitting in it |
| LEO-instruct 3D dialogue | 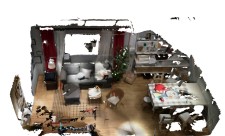 | USER: I'm redecorating my living room. Can you give me some suggestions on how to arrange the furniture? ASSISTANT: Of course! I'd be happy to help. What furniture do you have in your living room? USER: I have a sofa, a table, a chair, and a couch table. ASSISTANT: | Great! Here's a suggestion: you can place the sofa against the wall, with the table in front of it. The chair can be placed to the right of the table, and the couch table can be placed to the right of the chair. This arrangement creates a cozy seating area. |
| LEO-instruct task planning | 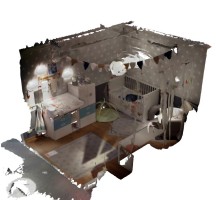 | USER: Develop a plan for this: Arrange and embellish the family room. ASSISTANT: | 1. Arrange and fluff the cushions and pillows on the armchair. 2. Place decorations, such as figurines or vases, on the cabinet or shelf. 3. Place hygiene products, such as wipes or lotions, on the commode. 4. Turn on the lights and adjust their brightness if needed. 5. Arrange diapers in a designated storage area. 6. Place a blanket or pillow on the armchair for added comfort. 7. Adjust the festoon or other decorative elements if desired. |

Table 26: **Examples from our datasets**. Continued.

| LEO-instruct embodied navigation |  | USER: The task is navigation. Your goal is to find counter by moving around in the scene. Past actions: <31999> <31999> <31999> <31999>. ASSISTANT: | <31996> |
| --- | --- | --- | --- |
| LEO-instruct robotic manipulation |  | USER: The task is manipulation. Your goal is to put the blue blocks in a green bowl. Past actions: <31991> <31671> <31511> <31991> <31671> <31511> <31991> <31671> <31511> <31991> <31671> <31511> <31991> <31671> <31511> <31991> <31671> <31511> <31991> <31671> <31511> <31991> <31671> <31511>. ASSISTANT: | <31748> <31644> <31511> <31736> <31595> <31500> |

