# OpenReview forum: "An Embodied Generalist Agent in 3D World"
_ICLR.cc/2024/Conference — Submitted to ICLR 2024_

### Official Review · Reviewer_bU9n · 2023-10-25

**Soundness:** 2 fair
**Presentation:** 4 excellent
**Contribution:** 3 good
**Rating:** 5
**Confidence:** 3

**Summary:**

This paper introduces LEO, a method that distinguishes itself from other LLM-based generalist agents by being able to understand and interact with the 3D world. LEO is trained with a two-stage process called "LEO-Align" and "LEO-Instruct". In the first stage, LEO is trained to align 3D vision and language by captioning 3D inputs (objects, objects-in-the-scene, and scenes). In the second stage, LEO is fine-tuned to specific tasks. LEO is demonstrated across a large variety of tasks including 3D vision-language understanding and embodied tasks. The authors provide detailed quantitative results of their method across multiple tasks: Scan2Cap, ScanQA, SQA3D, CLIPort, and ObjNav. The authors additionally ablate LEO to study components of their method including pretraining alignment, scale, and LEO's multitask nature.

**Strengths:**

- Overall this manuscript is very cleanly written, with well-presented figures.
- The results relating to 3D vision-language understanding and the corresponding ablations are compelling and informative.
- This work contains a novel and useful research direction, incorporating 3D understanding, LLMs, and embodied actions into a single model.
- This paper does a good job of providing quantitative and qualitative results for LEO across a broad range of tasks.

**Weaknesses:**

Overall the weaknesses in this work are found concerning the embodied actions tasks.
- In the CLIPort experiment the authors only demonstrate results on 3 out of the 10 tasks from the original experiment. I would be curious to see the full results of the left-out tasks to have a complete comparison to the original baselines in this experiment. This full table should also be presented in the supplementary information.
- The authors do not present any baselines to compare to for the embodied navigation task. The results that LEO obtains on the embodied navigation task are quite low when compared to other works in the embodied AI community. By just performing behavior cloning on 70k human demos, Habitat-Web [1] achieved a success rate of 27.8 on their MP3D test split. Furthermore, in PONI [2] the authors have a baseline called "BC" (Behavior Cloning) that is similar to LEO in modalities used and with the use of behavior cloning; however BC just utilizes a simple ResNet-50 architecture. This baseline performs comparably to LEO (3.8 Success vs 2.6/3.7 Success) on MP3D. Could the authors compare to previous work and comment on the "BC" baseline from PONI?

[1] Ramrakhya, Ram, et al. "Habitat-web: Learning embodied object-search strategies from human demonstrations at scale." Proceedings of the IEEE/CVF Conference on Computer Vision and Pattern Recognition. 2022.

[2] Ramakrishnan, Santhosh Kumar, et al. "Poni: Potential functions for objectgoal navigation with interaction-free learning." Proceedings of the IEEE/CVF Conference on Computer Vision and Pattern Recognition. 2022.

**Questions:**

Major questions and comments are given in the weaknesses section.
Minor questions and comments follow.
- In Habitat-Web the authors found a large difference in imitation learning results when trained on shortest path trajectories (4.4 success) and human demonstrations (35.4 success). The authors in this work chose to use the shortest path trajectories as they were less noisy and were easier to learn. Can the authors clarify if training on the human demonstrations led to worse quantitative performance on the task?
- In section 4.3 the authors claim they will present the soft-spl, but this is missing from Table 4.

---

> ### Author Response · Authors · 2023-11-20
> **Response to Reviewer bU9n (1/N)**
>
> We sincerely thank you for your time and constructive comments. Below, we provide detailed replies to your comments and hope we can resolve your major concerns.
>
> > Overall the weaknesses in this work are found concerning the embodied actions tasks.
> >
> > • In the CLIPort experiment the authors only demonstrate results on 3 out of the 10 tasks...This full table should also be presented in the supplementary information.
>
> We thank you for your comments. We only tested LEO on a subset of the original CLIPort tasks mainly due to storage resource constraints (the whole dataset is substantially larger than the original CLIPort tasks as we introduce object-centric 3D input). Scaling to all 10 tasks is definitely possible, but the storage cost will be huge while the evaluation is still IID. Therefore, per your suggestion, we’re in the process of scaling to more tasks but in a different route. Specifically, we are looking into replacing the current training tasks with open-ended ones using some recent work on LLM-based task generation [1], and performing zero-shot evaluation on the original CLIPort tasks. We believe increasing the diversity of tasks could help with better insight into the generalist agent architecture and training scheme of LEO in terms of zero-shot generalization. We commit to shipping these results ASAP and including them in the final version.

---

> > ### Author Response · Authors · 2023-11-20
> > **Response to Reviewer bU9n (2/N)**
> >
> > > • The authors do not present any baselines to compare...Could the authors compare to previous work and comment on the "BC" baseline from PONI?
> > > • In Habitat-Web the authors found a large difference in imitation learning results...Can the authors clarify if training on the human demonstrations led to worse quantitative performance on the task?
> >
> > Thank you for pointing this out! We’ve spotted some major issues in our objnav evaluation protocol since the submission of our manuscript. Therefore, we’ve re-worked the whole objnav experiment and that part in both sec. 4.3 and the appendix has been completely renovated. Here are some key insights:
> >
> > 1. We’ve switched to using the standard habitat MP3D objnav validation set (MP3D-val) and the protocol available in habitat-lab (https://github.com/facebookresearch/habitat-lab). Therefore, comparing with baselines like Habitat-web becomes possible. Additionally, we also evaluate LEO on the newly introduced HM3D objnav validation set (HM3D-val) (note: this is “zero-shot” as only MP3D scenes are used for training LEO) and compare it with a strong baseline from cortexbench [2]. For your convenience, we’ve posted the results below.
> >
> >     |  | MP3D-val |  | HM3D-val |  |
> >     | --- | --- | --- | --- | --- |
> >     | Model | Success (↑) | SPL (↑) | Success (↑) | SPL (↑) |
> >     | H.w. (shortest) | 4.4 | 2.2 | - | - |
> >     | H.w. (70k demo) | 35.4 | 10.2 | - | - |
> >     | VC-1 (ViT-B) | - | - | 57.1 | 31.4 |
> >     | LEO | 23.1 | 15.2 | 23.1 | 19.1 |
> >
> >     We would like to summarize the findings below:
> >
> >     - First of all, after resolving this evaluation issue, LEO is in fact producing **reasonable** performances on both standard MP3D-val and even HM3D-val in a zero-shot fashion.
> >     - Thanks to the 3D branch within LEO that processes 3D input and provides global 3D information(potentially offering coarse map details), LEO is able to attain better SPL than Habitat-web baseline, meaning that LEO is able to take shorter paths by leveraging the map information.
> >     - Compared to VC-1, an agent trained on HM3D, LEO offers reasonable results given the fact that it has never been trained on HM3D scenes.
> >
> > 2. Per your concern on LEO trained with human demonstrations vs. shortest path, we provide the following results in the appendix:
> >
> >     |  | MP3D-seen |  | MP3D-unseen |  |
> >     | --- | --- | --- | --- | --- |
> >     | Model | Success (↑) | SPL (↑) | Success (↑) | SPL (↑) |
> >     | Habitat-web (shortest) | 4.4 | 2.2 | - | - |
> >     | Habitat-web (70k demo) | 35.4 | 10.2 | - | - |
> >     | LEO (shortest) | 23.1 | 15.2 | 11.1 | 9.6 |
> >     | LEO (70k demo) | 7.1 | 5.3 | 8.9 | 8.6 |
> >
> >     As you can see, training with human demonstrations indeed leads to much worse performances of LEO and we believe the root cause is intuitive:
> >
> >     - LEO is able to process the extra global 3D scene input (a list of objects in the scene, including their point cloud and bbox information), therefore, learning with shortest path data makes sense — LEO could learn to convert the 3D scene input into map details, then learn to navigate to the target in shortest path by learning from the shortest path navigation data.
> >     - Further, as in our updated sec. 4.3 and H.2, LEO indeed **does not have any recurrent structure** when it comes to action prediction, i.e. the only thing relayed from the past is an action history of 4 — LEO effectively can be viewed as a transformer-based feed-forward policy, similar to RT-2 [3]. Therefore learning from human demonstration could be extra difficult, as it includes exploration, which can be hard to learn without a recurrent structure like RNN.
> >     - Note that, the choice of not having a recurrent structure is a result of the computation efficiency and scalability vs. performance tradeoffs. We commit to exploring more on this in future work.
> >
> > > • In section 4.3 the authors claim they will present the soft-spl, but this is missing from Table 4.
> > >
> >
> > We thank you for pointing this out. We believe this is a typo and has been fixed. Following the seminal works, we’ve decided to report success rate and SPL only, which should be sufficient to back our findings.
> >
> > [1] https://arxiv.org/abs/2310.01361
> >
> > [2] https://arxiv.org/abs/2303.18240
> >
> > [3] https://arxiv.org/abs/2307.15818
> >
> > We hope the above response can resolve your questions and concerns. Please let us know if there is any further question!

---

> ### Author Response · Authors · 2023-11-22
> **Do our responses address your concerns? Thank you again for your valuable feedback!**
>
> Dear Reviewer bU9n,
>
> We encourage you to view the demo video and animations we have provided at https://generalist-leo-anonymous.github.io. These resources offer a clearer understanding of the tasks we have benchmarked and the design of our model. We also added more visualizations of our collected data regarding at [this URL](https://docs.google.com/document/d/10KawUDUWWvSYwEVMDuwYMOYcwsdtxL97Teu87-5lZLs/edit?usp=sharing).
>
> **As we are approaching the end of the author-reviewer discussion phase, we want to check if our responses address your concerns well. Are there any further clarifications we could provide? We look forward to your feedback. Please feel free to let us know if we have missed any issues.**
>
> Thank you again for your valuable feedback!
>
> Best,
>
> Authors

---

### Official Review · Reviewer_Vn42 · 2023-10-30

**Soundness:** 2 fair
**Presentation:** 2 fair
**Contribution:** 2 fair
**Rating:** 3
**Confidence:** 4

**Summary:**

The paper identifies a gap in large language models (LLMs) regarding their limited capability to understand and interact with the 3D world. To address this, the authors introduce LEO, an embodied multi-modal and multi-task agent adept at tasks in the 3D environment. LEO is trained in two stages: 3D vision-language alignment and 3D vision-language-action instruction tuning, supported by a vast dataset containing object and scene-level multi-modal tasks. Through comprehensive testing, LEO excels in tasks like 3D captioning, embodied reasoning, and robotic manipulation. The paper also shares findings that can guide the development of future embodied generalist agents.

**Strengths:**

1.	The essence of the motivation is sound.
2.	The proposed pipeline is somewhat novel.
3.	The experiments cover many aspects. A lot of efforts have been made.

**Weaknesses:**

1. On the motivation and coherence: I think there are some word accuracy issues in the motivation.

(1)	In Intro - first paragraph: “This limitation … executing real-world tasks and the attainment of general intelligence.” First, many works on generalist agents are able to execute in the real world. (e.g. GATO, [2205.06175] A Generalist Agent (arxiv.org); PALM-E, [2303.03378] PaLM-E: An Embodied Multimodal Language Model (arxiv.org). To note, I do not ask for a baseline comparison against these works, just to suggest a better description of this work’s scope) The author should be more precise on what kinds of real-world tasks they cannot execute, and what kind of real-world tasks the paper plans to address. Second, about the attainment of general intelligence. How to define the scope of general intelligence? When the generalist agent is equipped with 3D understanding ability, can it attain general intelligence? I hope the author can narrow the scope of the statement by adding some transition sentences.

(2)	In Intro – second paragraph: the three challenges, the creation of datasets, design of scalable models, and effective learning strategies. I don’t see how it’s relevant to the following solutions. (a) The dataset is fused with existing datasets with high-quality data prompted by the LLMs. How is that challenging? With all the assets and models existing and known before. (b) The scalability of the model is not adequately validated. I understand that training a model from 7B parameter to 13B is already quite demanding for most labs, but it is far from emerging scaling effects. (c) the training strategy in LEO is nothing special. Again, if the known techniques can do well, how is that challenging?

2. On the novelty of the method: quite limited. To my understanding, it is a combination of all known techniques including tokenization, token embedding & LLM, and training & inference. Since they all “follow” some previous works. Besides, as a multi-task learning model, or “generalist agent” as the author may prefer, how the output looks is also important, but it is briefly described in the main text, and scattered in many places all over the paper.

3. On the novelty of dataset generation: it has more creative thoughts, but it is a pity to put too many details in the Appendix. I would suggest putting more of the method in the Appendix instead of the dataset generation details.

**Questions:**

What this paper puzzles me most is that it does not reveal the challenges and necessity of incorporating 3D input. In the response, I would like to see:

1.	What’s the particular technique to adopt to deal with the 3D input? How is it different from previous methods? Or when the previous method is combined, does it just work like that, or does something non-trivial happen?

2.	in this framework, according to the experiments, what ability requires 3D understanding to make it from 0 to 1 or improve a large margin? I think the author should focus on that, instead of trying to propose a “generalist agent”, which is exhaustive for a lab-level resource and really cannot produce any insightful outcomes. If the authors deem to the generalist agent story, then they should in-depth reveal the challenges, for example:

(1) how much data does it need, the key to creating and preparing the dataset?

(2) does the model parameter size matter at all, if 7B is already saturated (Fig 4-b) how about smaller models?

---

> ### Author Response · Authors · 2023-11-20
> **Response to Reviewer Vn42 (1/N)**
>
> We sincerely thank you for your time and constructive comments. Below, we provide detailed replies to your comments and hope we can resolve your major concerns.
>
> > 1. On the motivation and coherence: I think there are some word accuracy issues in the motivation.
> > ...First, many works on generalist agents are able to execute in the real world. (e.g. GATO...PALM-E...)...The author should be more precise on what kinds of real-world tasks they cannot execute, and what kind of real-world tasks the paper plans to address.
>
> Thanks for pointing this put. We agree that statements should be delivered carefully. We have clarified the background before this statement and here it is:
>
> ```
> Nonetheless, their abilities are primarily demonstrated within 2D domains, thereby limiting the comprehension of the 3D physical environment that envelops humans and other intelligent species.
> ```
>
> Our main point here is, prior works lack the capacity for 3D understanding, e.g., reasoning in complex 3D scenes, and performing scene-grounded dialogue with humans. To address these tasks in the real 3D world, we believe the 2D image input is insufficient (evidenced by [1,2,3,4]) and therefore equip the agent with 3D understanding ability. Specifically, all the 3D tasks (ex. 3D dialogue, 3D captioning[2], 3D QA[3], embodied reasoning [4]) LEO is evaluated on requires 3D input, and these tasks are indeed very crucial for embodied agents in the 3D world. Even the manipulation and navigation tasks, which by convention are mostly done with 2D input only, could enjoy some benefits when 3D is also taken into consideration (see the advantages of LEO over baselines on CLIPort tasks).
>
> > …Second, about the attainment of general intelligence...the author can narrow the scope of the statement by adding some transition sentences.
>
> Thanks for raising this in-depth question. This paper does not aim at defining what is general intelligence. Rather, our goal is to introduce a generalist agent that can perceive, ground, reason, plan, and act in the 3D world, which might but not necessarily, be an initial step towards more generally capable agents, or general intelligence to some extent.
>
> On the other hand, we never mean that the generalist agent can attain general intelligence as long as it is equipped with 3D understanding ability. Instead, we argue that the 3D understanding ability is a crucial component of embodied generalist agents in the 3D world, our evaluation of tasks including 3D dialogue, 3D captioning[2], 3D QA[3], and embodied reasoning [4], which are all quite relevant and important to embodied agents, has demonstrated exactly that.
>
> We've revised the relevant section to make this more clear as you suggested. Thanks again.
>
> > ...a) The dataset is fused with existing datasets with high-quality data prompted by the LLMs. How is that challenging?...
>
> Though the assets and LLMs exist, how to harness the LLMs to generate reliable 3D VL data is non-trivial and rarely explored. Text-only instruction-tuning data developed rapidly since Self-Instruct [5] and so did 2D VL instruction-tuning data since LLaVA [6]. In contrast, 3D VL instruction-tuning data lags behind. This is because representing 3D scenes with the LLM-familiar text is troublesome, e.g., tedious descriptions, using 3D bbox may pose great challenges due to excessive numerical reasoning for LLMs. There are also hallucination issues inherited from LLMs. To address the above problems, we propose a novel pipeline including scene-graph-based prompting and refinement procedures, which are non-trivial. Indeed, prompting LLMs into the responses we need for 3D-langauge data is not as easy as it might seem to be. Reviewer ADo8 even raises fundamental concerns on whether this is possible -- just to showcase how divergent our view on what LLMs can do today. The fact is, it is very capable, but not that much, leaving space for us to propose the aforementioned pipeline to make it work better in terms of data creation.

---

> > ### Author Response · Authors · 2023-11-20
> > **Response to Reviewer Vn42 (2/N)**
> >
> > > ...b) The scalability of the model is not adequately validated. I understand that training a model from the 7B parameter to 13B is already quite demanding for most labs, but it is far from emerging scaling effects.
> >
> > Thanks for your feedback. We've revised this claim, changing it from the design of scalable models to unified models, as a simple unified model for 3D embodied agents encompassing vision-language-action capabilities is our primary contribution.
> >
> > But despite that, we just would like to share with you some additional exploration on the scalability here. We do not obtain the expected results with larger models. When we scaled the LLM (Vicuna) from 7B to 13B, we were surprised that 13B was worse than 7B:
> >
> > |  | Scan2Cap | ScanQA | SQA3D | 3RQA |
> > | --- | --- | --- | --- | --- |
> > | LEO-NoAlign-NoAct-7B | 48.5 | 34.8 | 51.9 | 48.8 |
> > | LEO-NoAlign-NoAct-13B | 44.8 | 25.5 | 47.8 | 43.5 |
> >
> > Some further investigation is definitely of interest. But as we just mentioned above, we've revised our claim and scalability is not the focus on this work. We're happy to explore it in the near future.
> >
> > > ...c) the training strategy in LEO is nothing special. Again, if the known techniques can do well, how is that challenging?
> >
> > There are indeed many underexplored factors in the learning of such a 3D generalist agent, e.g., the data configurations and tuning settings. Though there are rich experiences with current 2D instruction-tuning strategies, how to seamlessly bridge the gap between 3D scene representation and LLM is rarely explored and pertains to significant challenges. Specifically, here are some major ventures of ours:
> > 1. We propose to learn 3D object-level referring and scene-level captioning as the alignment objectives.
> > 2. We adapt LLM to 3D scenes with LoRA techniques.
> >
> > Although both them might not be "completely novel", we believe they are new under this specific context of creating embodied generalist agents -- existing work, including RT-2[7] and GATO[12], did not use object-centric 3D input nor LoRA (GATO has trained everything from scratch, RT-2 simply fine-tune all parameters of a VLM, and both of them only use 2D input). **In our case, simply following these prior arts does not work**, as they do not offer a mechanism for processing 3D data, and are also not efficient to train (training from scratch or fine-tuning all parameters). Therefore, we believe the problem we are trying to solve (create a unified agent that can perform understanding, reasoning, and embodied action in a 3D world) is very challenging and our proposed solution makes a good contribution to the community.
> >
> >
> > > 2. On the novelty of the method: quite limited.
> >
> > We respectfully disagree with your assessment. Despite the similar framework with RT-2 [7], we aim at bridging the gap between 3D scene representation and LLM. We highlight some key challenges in the context of grounded 3D scene understanding, e.g., point cloud encoding, spatial relation modeling, modality alignment, and grounding. Existing efforts in addressing these challenges include ViL3DRel [8], 3D-VisTA [9], OpenScene [10], LERF [11], etc. How to design a simple unified model for grounded 3D scene understanding with the potential of supporting the learning of a more powerful 3D generalist agent is an unexplored question. To our knowledge, we are the first to build a 3D VLA generalist agent by injecting object-centric point clouds into LLMs.
> >
> > > Besides, as a multi-task learning model...how the output looks is also important, but it is briefly described in the main text, and scattered in many places all over the paper.
> >
> > Thanks for pointing this out. We illustrate the output format in our framework overview (Figure 1). The output tokens from LLM pertain to the LLM’s vocabulary and are further de-tokenized to text or actions. Since this is intuitive, we leave more space for the depiction of the components of our model.
> >
> > > 3. On the novelty of dataset generation: it has more creative thoughts, but it is a pity to put too many details in the Appendix. I would suggest putting more of the method in the Appendix instead of the dataset generation details.
> >
> > Thanks for your recognition of our creative thoughts in data collection. We have to compromise due to the space limit but we will try our best to bring the essential details back to the main text in the final version. Sorry about that!
> >
> > In the meantime, if you have any certain questions on data collection, feel free to ping us directly and we would love to help out!

---

> > ### Comment · Reviewer_Vn42 · 2023-11-21
> > **Reply to part 1**
> >
> > Thanks for your response. For this part:
> > First of all, it is not that if a model's input is a monocular RGB image, the model is "their abilities are primarily demonstrated within 2D domains". It is really a complex topic concerning 2D-3D, that's why I think the writing should be more careful. Let me develop the arguments:
> > 1. Many techniques can reconstruct 3D information from pure 2D images. Sometimes, they can skip the explicit reconstruction step and leverage the 3D information implicitly. RGB-based SLAM, and NeRF are among these lines of work. Since LEO's 3D ability is based on Global 3D, in a way, we can think methods like SLAM and NeRF are already performed. Thus, I see no reason why a pure 2D image-based pipeline cannot have the ability to deal with 3D.
> > 2. Both "ability" and "2D domains" are vague. PALM-E can conduct manipulation tasks in the 3D world. How can its ability be constrained in 2D domains? If the ability means the model's ability to process the input, not the ability of the whole system, it would make a lot more sense. However, since it comments on a "generalist agent", I would assume it is meant for the whole system. As for the 2D domain. Does it specifically mean the 2D input? or the task's workspace? Some tasks are performed in the 3D world, but its workspace is intrinsically 2D, like top-down grasping or top-down navigation. If it means the former, then I have argument 1. If it means for the latter, then it will make a lot more sense. But most tasks benchmarked in this paper are also intrinsically 2D.
> >
> > Second, I also recognize that the dataset creation has some good thoughts in it. But I just don't see any specifically in the main paper in Sec. 3. The writing makes me feel like "we do the text labeling with LLMs, but it is common practice, so we leave it to the appendix." If it is indeed non-trivial, then do not describe it like that. A major problem for me is when I read through the paper, I don't feel that dataset building and task execution are challenging.

---

> > > ### Author Response · Authors · 2023-11-22
> > > **Response to Reviewer Vn42 (Reply to rely1)**
> > >
> > > Dear Reviewer Vn42,
> > >
> > > Thank you for the response, we appreciate your comments and here are our opinions.
> > >
> > > > 1. Many techniques can reconstruct 3D information from pure 2D images… Thus, I see no reason why a pure 2D image-based pipeline cannot have the ability to deal with 3D.
> > > >
> > >
> > > "The assessment of a model's capability for 3D understanding requires a thorough and comprehensive evaluation, not just the ability to perform specific tasks or operate within particular scenarios. While PaLM-E demonstrates the capacity to manipulate objects in a 3D world, this alone does not equate to a robust understanding of grounded 3D scenes. We encourage examining a series of seminal 3D Vision-Language (VL) works such as ScanRefer [1], Scan2Cap [2], ScanQA [3], and SQA3D [4]. These benchmarks demonstrate a deeper, more holistic understanding of global 3D scenes, which is vital for any agent's proficiency in this domain. However, existing generalist models fall short in this regard.
> > >
> > > The cited works provide extensive experimental evidence, clearly indicating that reliance solely on 2D inputs, such as multi-view images, is inadequate for navigating the complexities inherent in 3D environments. This includes challenges like interpreting cluttered spaces and understanding spatial relationships. For instance, an experiment from Scan2Cap [2] (Table 3 in the original paper) comparing 2D-based methods against 3D-based approaches, vividly illustrating the significant difficulties in grasping spatial relations from 2D inputs.
> > >
> > > **Given these insights from the 3D VL community, it is our firm belief that a future generalist agent should incorporate a robust 3D understanding branch or module. Without it, how can these agents effectively navigate, reason, and perform question answering in 3D environments? While there might be more innovative methods to construct such a module, possibly by combining 2D views with reconstruction techniques or implicit representations, the exact approach remains an open question. It is essential to clarify that our stance is not to advocate for the exclusive use of 3D point cloud encoders as the sole solution. Our argument is for the undeniable necessity and inclusion of a proficient 3D understanding component within the architecture of future generalist agents.**
> > >
> > > |              | Acc (Category) | Acc (Attribute) | Acc (Relation) |
> > > |--------------|----------------|-----------------|----------------|
> > > | Oracle2Cap2D | 69.00          | 67.42           | 37.00          |
> > > | Oracle2Cap3D | 85.15 **(+16.15)**       | 72.22 **(+4.80)**            | 76.24 **(+39.24)**           |
> > >
> > > > 2. Both "ability" and "2D domains" are vague. PALM-E can conduct manipulation tasks in the 3D world. How can its ability be constrained in 2D domains?… But most tasks benchmarked in this paper are also intrinsically 2D.
> > > >
> > >
> > > We respectfully disagree with your characterization of the 'workspace' concept. The notion that an agent's navigation on a 2D floor or manipulation within a 2D table-top environment negates the need for extensive 3D knowledge is a misconception. Completing such tasks demands a deep understanding of 3D elements, including the geometric features of objects, spatial relationships, and precise referencing. These are inherently 3D tasks, and while methods like PALM-E may approach them using image-based 2D representations, this approach falls short of providing the comprehensive 3D capabilities necessary for a holistic understanding of 3D environments. Such understanding is critical for enabling an agent to reason, answer questions, and navigate effectively in these spaces. Echoing the tasks we benchmarked in the paper, such as ScanRefer[1], ScanQA[3], SQA[4], and embodied navigation [17].
> > >
> > > **Furthermore, and in alignment with our earlier response, the tasks described in our paper are intrinsically 3D in nature. To demonstrate this, we have provided visualizations and demonstrations at an anonymous website https://generalist-leo-anonymous.github.io. We strongly encourage you to review these materials for a clearer, more direct illustration of how our tasks fundamentally require 3D understanding and significantly expand the spectrum of tasks considered in previous generalist agent research.**
> > >
> > > Your insights are valuable, but we firmly believe that our approach and understanding of 3D tasks and environments are crucial for advancing the field. We look forward to further discussions on this matter.

---

> > > > ### Author Response · Authors · 2023-11-22
> > > > **Response to Reviewer Vn42 (Reply to rely1, cont'd)**
> > > >
> > > > > Second, I also recognize that the dataset creation has some good thoughts in it. But I just don't see any specifically in the main paper in Sec. 3… A major problem for me is when I read through the paper, I don't feel that dataset building and task execution are challenging.
> > > > >
> > > >
> > > > Thank you for recognizing the efforts we have invested in creating our dataset. We are glad that you appreciate the thoroughness of our data collection process. As you pointed out, there are indeed numerous intricate details involved in this endeavor. Due to the constraints of page limits, we opted to include these specifics in the appendix. This decision was made to ensure the paper remains focused and concise, especially since our primary aim is to highlight the construction of our proposed embodied agent.
> > > >
> > > > We understand that the collection of the dataset involved extensive work and detailed analysis. To ensure transparency and facilitate further research, we plan to open-source the dataset. This will allow interested readers to delve into these details at their discretion.
> > > >
> > > > Nonetheless, we want to underscore the considerable challenges in prompting 3D scene-grounded data. And our collected high-quality data can fill the blank in 3D VL commuity. Check more examples at [this url](https://docs.google.com/document/d/10KawUDUWWvSYwEVMDuwYMOYcwsdtxL97Teu87-5lZLs/edit?pli=1).
> > > >
> > > > **In accordance with your valuable suggestion, we will refine our paper. We aim to more effectively highlight the challenges we encountered and our contributions in the dataset section. This will ensure a balanced presentation of both the dataset creation and the construction of our embodied agent.**
> > > >
> > > > Thank you once again for your constructive feedback, which we believe will greatly enhance the quality of our work.

---

> ### Author Response · Authors · 2023-11-20
> **Response to Reviewer Vn42 (3/N)**
>
> > 1. What’s the particular technique to adopt to deal with the 3D input? How is it different from previous methods? Or when the previous method is combined, does it just work like that, or does something non-trivial happen?
>
> We adopt a point cloud encoder to process each object-centric point cloud into a per-object feature (embedding), we call this "tokenization" over 3D input. A subsequent Spatial Transformer [8] applies spatial attention based on the object features and object locations to reason about the spatial relations in the 3D scenes. Different from 3D-LLM [15], which constructs a complicated pipeline that leverages various 2D foundation models to fuse 3D scene representation. We explore to design a simple unified model to feed the 3D scenes into LLMs.
>
> All these details can be found in both sec. 2.2 and appendix D.2. Feel free to let us know if there is anything you would like to learn about about!
>
> > 2. in this framework, according to the experiments, what ability requires 3D understanding to make it from 0 to 1 or improve a large margin?
>
> Thanks for pointing this out. Here are the abilities:
>
> 1. All the 3D-language understanding and reasoning capacity, including 3D dialogue, 3D captioning [2], 3D QA [3], embodied reasoning [4], etc. These tasks not only requires scene-level 3D input (only 2D input won't work), but also are incredibly relevant to building a embodied generalist agent that can perceive, reason and act in a 3D world.
>
> 2. Embodied AI capacity, including navigation and manipulation.
>
>     - For the navigation task, in our re-worked ObjNav experiment (please find more details in our additional response in the end), LEO, which effectively adopts a Markvian policy and learn from "shortest path" data, is able to achieve much better performances than baselines that uses 2D only and the same data. The intuition is the global 3D input, which potentially offers map details, allows LEO to think globally and plan shortest path towards the target object without the need for recurrent structure like RNN.
>
>     - For the manipulation tasks, LEO is able to attain better performances, especially on unseen splits of the CLIPort tasks we evaluated, through the use of additional 3D input.
>
> > (1) how much data does it need, the key to creating and preparing the dataset?
>
> Thank you for rasing this question. We've explored a "PartialData" configuration with 10% instruction-tuning data in our ablation study in sec. 4.4., which shows significantly worse results than the one tuned with full data. Although it is still hard to tell how much data does it need, but such result could shed some light on the scale of data LEO might needs.
>
> Just to elaborate a bit on data, previous works also indicate that the quality is more important than the quantity for instruction-tuning data [12,13,14]. In preparation of LEO, we've dedicated to developing several techniques on improving the data quality, including O-CoT(sec. 3.3), an pioneering approach to prompt LLM into generating high-quality 3D-language data. We will sure consider how to improve the instruction-tuning data quality further in future work.
>
> > (2) does the model parameter size matter at all, if 7B is already saturated (Fig 4-b) how about smaller models?
>
> Thanks for pointing this out. Exploring smaller models is indeed a great direction to explore. Point is, the smallest model that Vicuna (the LLM we adopted) could offer is 7b. But we commit to exploring some other LLM architectures with smaller scale, ex. T5, in the future work.

---

> > ### Comment · Reviewer_Vn42 · 2023-11-21
> > **Reply to part3**
> >
> > Thanks for your response!
> > Please don't take me wrong. I certainly know the technical differences from the previous works, but by breaking this technical difference down:
> > 1. Adopting a 3D encoder. An obvious choice, while is the same idea to process a 2D image.
> > 2. Then the question is what the 3D encoder should be, ok, a pre-trained point cloud encoder.
> > 3. Then, we need to encode 3D relations. Ok, how it's done? another previous work.
> >
> > All these technical choices are straightforward. It's like there's no challenge when making these choices.
> >
> > So I am more interested in: when you have a new kind of input, what new ability it can achieve?
> > If it is just a performance boost on the existing benchmarks, that's pretty expected, since the work inputs global 3D information.

---

> ### Author Response · Authors · 2023-11-20
> **Response to Reviewer Vn42 (4/N)**
>
> ---
> As a side note, we’ve spotted some major issues in our navigation (one task in embodied action) evaluation protocol since the submission of our manuscript. Therefore, we’ve re-worked the whole objnav experiment and that part in both sec. 4.3 and the appendix has been completely renovated. Here are some key insights:
>
> 1. We’ve switched to using the standard habitat MP3D objnav validation set (MP3D-val) and the protocol available in habitat-lab (https://github.com/facebookresearch/habitat-lab). Therefore, comparing with baselines like Habitat-web becomes possible. Additionally, we also evaluate LEO on the newly introduced HM3D objnav validation set (HM3D-val) (note: this is “zero-shot” as only MP3D scenes are used for training LEO) and compare it with a strong baseline from CortexBench [17]. For your convenience, we’ve posted the results below.
>
>     |  | MP3D-val |  | HM3D-val |  |
>     | --- | --- | --- | --- | --- |
>     | Model | Success (↑) | SPL (↑) | Success (↑) | SPL (↑) |
>     | H.w. (shortest) | 4.4 | 2.2 | - | - |
>     | H.w. (70k demo) | 35.4 | 10.2 | - | - |
>     | VC-1 (ViT-B) | - | - | 57.1 | 31.4 |
>     | LEO | 23.1 | 15.2 | 23.1 | 19.1 |
>
>     We would like to summarize the findings below:
>
>     - First of all, after resolving this evaluation issue, LEO is in fact producing **reasonable** performances on both standard MP3D-val and even HM3D-val in a zero-shot fashion.
>     - Thanks to the 3D branch within LEO that processes 3D input and provides global 3D information(potentially offering coarse map details), LEO is able to attain better SPL than Habitat-web baseline, meaning that LEO is able to take shorter paths by leveraging the map information.
>     - Compared to VC-1, an agent trained on HM3D, LEO offers reasonable results given the fact that it has never been trained on HM3D scenes.
>
> 2. Per your concern on LEO trained with human demonstrations vs. shortest path, we provide the following results in the appendix:
>
>     |  | MP3D-seen |  | MP3D-unseen |  |
>     | --- | --- | --- | --- | --- |
>     | Model | Success (↑) | SPL (↑) | Success (↑) | SPL (↑) |
>     | Habitat-web (shortest) | 4.4 | 2.2 | - | - |
>     | Habitat-web (70k demo) | 35.4 | 10.2 | - | - |
>     | LEO (shortest) | 23.1 | 15.2 | 11.1 | 9.6 |
>     | LEO (70k demo) | 7.1 | 5.3 | 8.9 | 8.6 |
>
>     As you can see, training with human demonstrations indeed leads to much worse performances of LEO and we believe the root cause is intuitive:
>
>     - LEO is able to process the extra global 3D scene input (a list of objects in the scene, including their point cloud and bbox information), therefore, learning with shortest path data makes sense — LEO could learn to convert the 3D scene input into map details, then learn to navigate to the target in shortest path by learning from the shortest path navigation data.
>     - Further, as in our updated sec. 4.3 and H.2, LEO indeed **does not have any recurrent structure** when it comes to action prediction, i.e. the only thing relayed from the past is an action history of 4 — LEO effectively can be viewed as a transformer-based feed-forward policy, similar to RT-2 [7]. Therefore learning from human demonstration could be extra difficult, as it includes exploration, which can be hard to learn without a recurrent structure like RNN.
>     - Note that, the choice of not having a recurrent structure is a result of the computation efficiency and scalability vs. performance tradeoffs. We commit to exploring more on this in future work.

---

> ### Comment · Reviewer_Vn42 · 2023-11-21
> **Reply to part 2**
>
> Thanks for your response, especially the results with scaling models.
>
> For this part:
> I agree with you for the most part. I understand the contribution of filling the gap of one kind format of input-output or task setting, and I recognize the design of the pipeline as "novel" in the Strength part.
>
> But for the method itself, as a unified model for 3D generalist agent, yes, no method has done that before. But let's look at Fig 1, If I just remove the "3D encoder", it is nothing different from a 2D-based generalist agent. And the idea to encode a 3D input, turn it into a feature and concatenate it with the rest of the features, is not different from the 3D referring expression task. There exist no technical contributions but just a new task setting.
> I won't argue if this new task setting with a static global 3D input is meaningful because different people have different opinions.
>
> That's why, to sum up, I think the method's novelty is limited.
>
> In this case, I will check the contribution on the dataset side, which I also find problematic. (mentioned in the main review and "reply to part1")

---

> > ### Author Response · Authors · 2023-11-22
> > **Response to Reviewer Vn42 (Reply to reply 2&3)**
> >
> > Thank you for your feedback and clarification. Since reply 2 and reply 3 both concern the novelty of our method, we address them together in this response.
> >
> > > But for the method itself, as a unified model for 3D generalist agent, yes, no method has done that before... That's why, to sum up, I think the method's novelty is limited
> >
> > We would like to clarify the scope of the 'method' as mentioned in your comment. This clarification is essential to accurately assess the novelty and contributions of our work.
> >
> > (1) If 'method' pertains to the model's architecture, then labeling its novelty as limited is a disservice not only to our work but also to a lineage of significant prior research. Take, for instance, Gato and PaLM-E, both employing a straightforward Transformer architecture to construct their pipelines. The techniques they utilize, such as encoding different modalities, though derived from previous works, are crucial in building more powerful models with new capabilities. The AI community values such incremental advancements. Therefore, we assert that the assessment of novelty should be more nuanced, acknowledging the collective progress in our field.
> > (2) Should 'method' indicates the entire pipeline design, then we shall emphasize the distinct novelty of our approach. Our method, encompassing a unified task sequence, a two-stage learning scheme, and an efficient tuning strategy with LoRA, represents a significant leap forward. As far as we are aware, ours is the pioneering work in creating a vision-language-action generalist agent adept in 3D scene understanding and interaction.
> >
> > > All these technical choices are straightforward. It's like there's no challenge when making these choices.
> >
> > Constructing LEO's architecture, while seemingly intuitive, involved complex and non-trivial decisions. Key challenges included aligning 3D scene and language representations effectively with LLMs, **which has not been achieved before**. Our work in designing the scene encoding module, such as optimizing 3D representation alignment and exploring point cloud encoders, was intricate. We also identified limitations in the Spatial Transformer, highlighting the complexities in bridging the gap between scene representation and Large Language Models (LLM).
> >
> > > So I am more interested in: when you have a new kind of input, what new ability it can achieve? If it is just a performance boost on the existing benchmarks, that's pretty expected, since the work inputs global 3D information.
> >
> > We appreciate your focus on the role of 3D vision in our LEO system and would like to clarify a crucial aspect regarding its implementation. It seems there might be a misunderstanding based on the expectations set from 2D agent capabilities. In the case of LEO, for most tasks, the 2D image encoder is not the primary component; instead, the 3D branch predominantly handles visual perception. The only exception is in embodied acting tasks, where 2D input is utilized for ego perception to facilitate interaction.
> >
> > Therefore, in 3D Vision-Language (VL) tasks, we do not introduce a 'new kind of input' as might be expected. However, to provide a clearer picture of the impact and effectiveness of the 3D vision branch, we have included a quantitative comparison in our study. This comparison shows the advantages and capabilities brought about by incorporating the 3D branch in LEO.
> >
> > | Tasks | Cap3D | SceneCap | Scan2Cap | ScanQA | SQA3D | 3RQA |
> > | --- | --- | --- | --- | --- | --- | --- |
> > | LEO (with 3D input only)| 35.9 | 65.5 | 64.9 | 36.8 | 54.1 | 61.3 |
> > | LEO (with no visual input) | 16.2 | 58.6 | 43.4 | 32.8 | 50.7 | 51.5 |
> >
> > In embodied acting tasks, we further ablate 2D and 3D branch to reveal their contributions respectively.
> >
> >
> > | MP3D-val | Success (↑) | SPL (↑) |
> > | --- | --- | --- |
> > | Habitat-web (shortest) | 4.4 | 2.2 |
> > | Habitat-web (70k demo) | 35.4 | 10.2 |
> > | LEO (w/o 3D) | 8.6 | 6.8 |
> > | LEO (w/o 2D) | 7.8 | 4.6 |
> > | LEO (full) | 23.1 | 15.2 |
> >
> > Your question about what ability may emerge when combining 2D and 3D together is inspiring but quite open, which yet falls out of the scope of our current paper.

---

> ### Author Response · Authors · 2023-11-22
> **Response to Reviewer Vn42 (5/N)**
>
> We first refine our references in this response. Our reply to your latest concerns comes in the following responses.
>
> [1] Dave Zhenyu Chen, et al. Scanrefer: 3d object localization in rgb-d scans using natural language. ECCV 2020.
>
> [2] Zhenyu Chen, et al. Scan2cap: Context-aware dense captioning in rgb-d scans. CVPR 2021.
>
> [3] Daichi Azuma, et al. Scanqa: 3d question answering for spatial scene understanding. CVPR 2022.
>
> [4] Xiaojian Ma, et al. Sqa3d: Situated question answering in 3d scenes. ICLR 2023.
>
> [5] Yizhong Wang, et al. Self-instruct: Aligning language model with self generated instructions. ACL 2023.
>
> [6] Haotian Liu, et al. Visual instruction tuning. NeurIPS 2023.
>
> [7] Anthony Brohan, et al. Rt-2: Vision-language-action models transfer web knowledge to robotic control. arXiv preprint arXiv:2307.15818, 2023.
>
> [8] Shizhe Chen, et al. Language conditioned spatial relation reasoning for 3d object grounding. NeurIPS 2022.
>
> [9] Ziyu Zhu, et al. 3d-vista: Pre-trained transformer for 3d vision and text alignment. ICCV 2023.
>
> [10] Songyou Peng, et al. Openscene: 3d scene understanding with open vocabularies. CVPR 2023.
>
> [11] Justin Kerr, et al. Lerf: Language embedded radiance fields. ICCV 2023.
>
> [12] Long Ouyang, et al. Training language models to follow instructions with human feedback. NeurIPS 2022.
>
> [13] Rohan Taori, et al. Stanford alpaca: An instruction-following llama model. https://github.com/tatsu-lab/stanford_alpaca, 2023.
>
> [14] Wei-Lin Chiang, et al. Vicuna: An open-source chatbot impressing gpt-4 with 90%* chatgpt quality. https://lmsys.org/blog/2023-03-30-vicuna, 2023.
>
> [15] Yining Hong, et al. 3d-llm: Injecting the 3d world into large language models. NeurIPS 2023.
>
> [16] Scott Reed, et al. A generalist agent. TMLR 2022.
>
> [17] Arjun Majumdar, et al. Where are we in the search for an artificial visual cortex for embodied intelligence? arXiv preprint arXiv:2303.18240, 2023.

---

> ### Author Response · Authors · 2023-11-22
> **It has been a delightful discussion and we sincerely hope you can increase the evaluation score**
>
> Dear Reviewer Vn42
>
> It has been a delightful experience to engage in a discussion over the technical details and contributions of our submission. We greatly appreciate your rich experience and insightful feedback in this field. To summarize, we would like to highlight several key points:
> 1. **Dataset Contribution Not Sufficiently Highlighted in the Main Paper**
> Thank you for the acknowledgment. We will polish the paper and emphasize the challenges, efforts, and contributions of building the datasets in a proper way.
>
> 2. **The Necessity of 3D for an Embodied Generalist Agent and Our Model's Evidence of This Requirement**
> We show the experimental results that demonstrate that 3D input is essential for our 3D tasks. Fundamentally, most of our tasks are 3D based and do not rely on image inputs. We encourage you to view the demo video and animations we have provided at https://generalist-leo-anonymous.github.io. These resources offer a clearer understanding of the tasks we have benchmarked and the design of our model. The question of whether to use images as inputs for constructing 3D representations is indeed an open one. However, it is important to note that this topic falls outside the scope of our current paper. Our focus is on illustrating the essential role of 3D inputs in the functionality of our embodied generalist agent.
>
> 3. **Concerns Over Limited Technical Contribution of the Method**
> We firmly believe that striving for a more capable embodied generalist represents a long-term aspiration within our field. Our research, hopefully the first in a series, aims to establish a foundational milestone for the development of generalist agents that can effectively comprehend the real 3D world. *It is not uncommon for there to be differing opinions regarding the novelty of scientific work, particularly in endeavors that seek to break new ground*. However, we wish to emphasize our substantial contributions in this area. Our work involves a new model designed and trained based on a large language model, carefully curating benchmarks, and achieving robust performances across established benchmarks over a large spectrum of 3D and embodied tasks. These contributions are distinctive and play a pivotal role in advancing the community's efforts toward creating more capable generalist agents with an enhanced understanding of the 3D world.
>
> **In light of these considerations, we respectfully request the reviewers to reconsider and potentially increase the evaluation score of our work. We deeply value your feedback and hope that our contributions are recognized as significant steps forward in this exciting and evolving field.**

---

### Official Review · Reviewer_foVx · 2023-10-31

**Soundness:** 4 excellent
**Presentation:** 4 excellent
**Contribution:** 4 excellent
**Rating:** 8
**Confidence:** 4

**Summary:**

In this paper, the authors propose an embodied multi-modal and multi-task generalist agent, aiming to improve the agent's ability to understand and interact with the 3D world. The experiments demonstrate the effectiveness of the proposed LEO under various embodied tasks.

**Strengths:**

1. The motivation of this paper is interesting and reasonable since the LLM-based agent needs the ability to complete 3D tasks for applying in the real world. I agree with the viewpoint that "advancements in 3D scene-level understanding have significantly lagged behind".

2. The LEO can perceive, ground, reason, plan, and act in the 3D world simultaneously and obtain promising results.

3. The dataset and fine-tuning method for constructing generalist LEO provides good contributions and insights for the embodied AI community.

4. Extensive details and demonstrations are provided in the appendix, which makes the work easy to follow, and experiments are sufficient.

**Weaknesses:**

- The section of related works should be included in the main text to ensure the completeness of the paper.

- The way to achieve vision-language-action simply follows RT-2, thus lacking some novel design and weakening the technical contribution of this work.

- The authors need to further polish the writing.

**Questions:**

- Why is the 3D point cloud data for the third-person global view needed by an embodied agent?

---

> ### Author Response · Authors · 2023-11-20
> **Response to Reviewer foVx**
>
> We sincerely thank you for your time and constructive comments. Below, we provide detailed replies to your comments and hope we can resolve your major concerns.
>
> > • The section of related works should be included in the main text to ensure the completeness of the paper.
>
> Thanks for pointing this out. We have to make a compromise due to the page limit for submission. We will try out best bring it back in the final version.
>
> > • The way to achieve vision-language-action simply follows RT-2, thus lacking some novel design and weakening the technical contribution of this work.
>
> RT-2 [1] inspires us on the action tokenization and we adopt a similar framework to RT-2. Nonetheless, we make several key contributions beyond RT-2:
> 1. We adapt such a framework to 3D scene understanding, and explore to bridge the gap between object-centric point clouds and human language, while GATO [6] and RT-2 [1] are both 2D only, without considering 3D information.
> 2. We seek to endow the agent with the capabilities of interacting with humans in open scenarios, e.g., dialogue, via 3D instruction tuning. Moreover, we also investigate the agent’s properties, e.g., instruction-tuning data ablation, and scaling law analysis.
> 3. Instead of training from scratch like GATO [6] or finetuning all parameters as in RT-2 [1], we have ventured into parameter-efficient tuning using LoRA, which has never been explored before in such vision-language-action agents and we believe our conclusions on how this training scheme scale as data and model size grow (see sec. 4.4/4.5) could make valuable contributions to the community.
>
> > • The authors need to further polish the writing.
>
> Thanks for your feedback on writing. We put lots of efforts in delivering a well-written paper and have polished our paper for the revised version.
>
> > Why is the 3D point cloud data for the third-person global view needed by an embodied agent?
>
> From a high level, providing global 3D input, regardless of being rarely explored in the current embodied AI tasks, is indeed quite popular in robots being deployed to the real world -- in fact, the popularity of mapping techniques like SLAM has demonstrated the need for obtaining 3D information. Therefore, we believe providing such information is useful for embodied agents.
>
> More specifically, in the case of the object navigation of LEO, since LEO has a Markovian policy without a recurrent structure (ex. RNN), obtaining global 3D information is crucial, as exploring the scene and finding objects from 2D input only requires RNN, while with global 3D input (potentially offering map details), LEO will be able to think globally and learn to find the shortest path towards the target object.
>
> For the other tasks, LEO demonstrated (3D dialogue, embodied reasoning with SQA3D [5], etc), all of them are effectively 3D tasks, therefore 3D input is definitely needed.
>
> We hope the above response can resolve your questions and concerns. Please let us know if there is any further question!
>
>
> [1] Anthony Brohan, et al. Rt-2: Vision-language-action models transfer web knowledge to robotic control. arXiv preprint arXiv:2307.15818, 2023.
>
> [2] Dave Zhenyu Chen, et al. Scanrefer: 3d object localization in rgb-d scans using natural language. ECCV 2020.
>
> [3] Zhenyu Chen, et al. Scan2cap: Context-aware dense captioning in rgb-d scans. CVPR 2021.
>
> [4] Daichi Azuma, et al. Scanqa: 3d question answering for spatial scene understanding. CVPR 2022.
>
> [5] Xiaojian Ma, et al. Sqa3d: Situated question answering in 3d scenes. ICLR 2023.
>
> [6] Scott Reed, et al. A generalist agent. TMLR 2022.

---

### Official Review · Reviewer_ADo8 · 2023-11-06

**Soundness:** 2 fair
**Presentation:** 2 fair
**Contribution:** 2 fair
**Rating:** 5
**Confidence:** 4

**Summary:**

The paper proposes an embodied multi-modal agent called LEO that can understand and interact with 3D world. LEO builds on top of Vicuna-13B large language model (LLM), and learns to encode 2D images as well as 3D point-cloud of the environment. The model is capable of generating natural language text (like caption, or answers to questions about the environment), as well as actions (for navigation or manipulation). To train this model, the authors collect a large 3D vision-language alignment dataset at object-level and scene-level to learn grounding. Further, this  model is fine-tuned on several tasks including captioning, question-answering, dialogue, planning, navigation and manipulation.

**Strengths:**

- The authors attempt a fairly ambitious idea of building generalist embodied agents which can not only understand 2D content, but also 3D scenes. The proposed approach aims to generate both text as well as actions. The effort to integrate all these signals into the model, and setting up training on so many tasks is commendable.
- The authors rightly point out that progress in building generalist embodied agent is severely restricted by large-scale ground-truth data. Collecting such data manually is expensive and restrictive so I agree with the overall idea (regardless of its execution in the paer) of collecting this data semi-automatically using LLMs.

**Weaknesses:**

1. I have strong concerns regarding automatic data-collection using LLM and its accuracy. For instance, in Table 21, the authors describe a single example from task-specific datasets. In that table:
    1. task planning describes “wiping away dust”, “checking the stability of chair”, “adjusting the height of the chair”, “vacuuming dirt or crumbs”, “checking the lighting”. This low-level plan is clearly not grounded in the scene nor the embodiment of the robot. The chairs depicted in the scene cannot be adjusted, how will the robot change the temperature / ventilation / lighting of the scene. This kind of pre-training / fine-tuning is unnecessary and incorrect.
    2. Similarly, the 3D captioning task example has “a white cabinet in the corner of the room. in the direction from the door and from the inside. it will be on the left, there is a small brown table on the left side of the cabinet and a smaller table on the right side of the cabinet” This example clearly has a lot of grammatical mistakes, and makes me question the GT data against which the system is evaluated.
    3. For scene captioning, the generated caption has very little overlap with the scene. It describes the chair as being of the same color as the floor (which is incorrect). Their 3D positions are described incorrectly as well.
2. The ablations performed in the paper are unclear and inadequate. The paper doesn’t perform any component level ablations on the model (removing 2D images as input, removing 3D images as input, checking against a blind baseline). Without such ablations, there is no way to tell how important each components of the system are. Additionally, it’s unclear what the take-aways message from ablations (Table 18, and Figure 4(a)) are.
3. It seems that a bunch of decisions in the paper (box-based vs scene-graph prompting, etc) are taken by qualitatively comparing the output. This is an insufficient way to justify design decisions (specialy for GT data collection), and without good-faith human evaluation / quantitative evaluation, it’s unclear if the GT can be trusted or not. This puts into question the entire GT dataset, and its unclear if any followup evaluation on this benchmark can be trusted or not. For instance, in Figure 13, it’s unclear which one of box-based prompting or scene-graph based prompting is better.  Similarly, in Table 11, the author uses GPT-4’s evaluation of two generated captions to decide that partial scene graph is sufficient. I think relying on GPT-4 to subjectively evaluate two captions and checking whether it’s sufficiently grounded to a 3D scene is problematic.
4. Results on object-navigation are surprisingly low. Can the authors comment on why is the success rate so low compared to state of the art object navigation methods. Additionally, the object navigation setup requires the agent to navigate “unexplored” environments. But the proposed approach assumes a static “known” environment. Can the authors please clarify the experimental setup for object navigation?

**Update**
The authors addressed some of my concerns. It was good to see updated experiments (for Object Navigation), and further ablations. However, I am not convinced about the data. While, I am aware that LLM based augmentation is very popular, I think the dataset quality is still pretty low. The explanation that these annotations build on top of existing datasets and hence suffer from the underlying data quality issues isn't sufficient. I also think that the blind baselines (presented during rebuttal) clearly are not very far off from visual baselines. This also shows that either the task is not challenging enough, or the gains coming from the proposed method are not big enough. Overall, I think the paper needs significant refinement and the authors will benefit from going back to the drawing board, and rethink the story and the presentation of their ideas. However, I am increasing my score a little bit to reflect my new thoughts.

**Questions:**

Apart from questions asked in the weakness section, here are some additional questions:

1. How does the agent handle multi-step trajectory of the agent (observation action pair for multiple time steps)? Is the agent only trained as a markovian policy which doesn’t require encoding the history? Or is the trajectory encoded by simply appending each observations sequentially into the LLM?
2.  CLIPort numbers in Table 3 are lower than the highest reported numbers in the CLIPort paper. For completeness and accuracy, can the authors update the CLIPort numbers to the highest numbers reported in the paper?

---

> ### Author Response · Authors · 2023-11-20
> **Response to Reviewer ADo8 (1/N)**
>
> We sincerely thank you for your time and constructive comments. Below, we provide detailed replies to your comments and hope we can resolve your major concerns.
>
> > I have strong concerns regarding automatic data-collection using LLM and its accuracy...
> > 1. task planning describes...This kind of pre-training / fine-tuning is unnecessary and incorrect.
> > 2. Similarly, the 3D captioning task...This example clearly has a lot of grammatical mistakes, and makes me question the GT data against which the system is evaluated.
> > 3. For scene captioning...Their 3D positions are described incorrectly as well.
>
> Thank you so much for pointing this out! Indeed, generating data using LLMs could be challenging, and that's why we meticulously design several refinement procedures to enhance the data quality (details shown in Appendix A.2, A.3 and A.4). Notably, considering 1) the inherently scarce 3D VL data, 2) the community’s explorations in prompting LLM [1,2,3,4,5,6,7], and 3) our improved scene-graph-based prompting approach, we believe our approach can provide high-quality 3D VL data and greatly contribute to the 3D VL community. We add more visualizations of our data regarding your concerns at [this URL](https://docs.google.com/document/d/10KawUDUWWvSYwEVMDuwYMOYcwsdtxL97Teu87-5lZLs/edit?usp=sharing).
>
> More specifically:
>
> 1. The data example of task planning in Table 21. You mentioned the chair is not adjustable and the robot cannot change the temperature etc. However, we remind that the role of LEO in this task is an AI assistant that first of all, produce plans that are based on the current scene layouts (ex. furnitures in the scenes), then communicate with human and offer suggestions. It never means to actually perform those tasks. Therefore, as long as the produced plan is fully grounded (which it does, the plan in Table 21 clearly includes chairs and tables that are in the scene), and reasonable for humans (that includes adjusting the temperature), it will be viewed as a good plan.
>
> 2. The 3D captioning data comes from Scan2Cap (ScanRefer) dataset, one of existing 3D VL datasets, which have some grammatical flaws not caused by us. Compared with these datasets, our own curated data features more natural textual phrases and are mostly grammatically correct thanks to ChatGPT. Check out more comparative examples [here](https://docs.google.com/document/d/10KawUDUWWvSYwEVMDuwYMOYcwsdtxL97Teu87-5lZLs/edit?usp=sharing).

---

> > ### Author Response · Authors · 2023-11-20
> > **Response to Reviewer ADo8 (2/N)**
> >
> > 3. Coming up, scene captioning. The generated captions are mostly induced from the scene graph. For the specific example you mentioned, the provided scene graph can be found below:
> >
> >     ```json
> >     'floor-1': {
> >         'relations': [], 'attribute': {'material': 'wooden', 'shape': 'flat', 'lexical': 'inside', 'color': 'brown'}
> >     },
> >     'wall-2': {
> >         'relations': ['attached to floor-1'], 'attribute': {'shape': 'flat', 'lexical': 'architectural', 'color': 'white'}
> >     },
> >     'wall-3': {
> >         'relations': ['attached to floor-1'], 'attribute': {'shape': 'flat', 'lexical': 'architectural', 'color': 'white'}
> >     },
> >     'table-4': {
> >         'relations': ['standing on floor-1', 'close by chair-5', 'to the left of chair-5', 'close by chair-6'],
> >         'attribute': {'material': 'wooden', 'shape': 'rectangular', 'other': 'rigid', 'state': 'tidy', 'color': 'brown', 'size': 'tall'}
> >     },
> >     'chair-5': {
> >         'relations': ['standing on floor-1', 'to the right of table-4', 'close by table-4', 'to the right of chair-6', 'behind chair-6'],
> >         'attribute': {'size': 'tall'}
> >     },
> >     'chair-6': {
> >         'relations': ['standing on floor-1', 'close by table-4', 'in front of chair-5', 'to the left of chair-5'],
> >         'attribute': {'other': 'rigid', 'size': 'tall'}
> >     },
> >     'chair-7': {
> >         'relations': [], 'attribute': {'other': 'rigid', 'size': 'tall'}
> >     },
> >     'wall-12': {
> >         'relations': [], 'attribute': {'lexical': 'architectural'}
> >     }
> >     ```
> >      And the response from ChatGPT, which is part of our dataset, is:
> >      ```
> >      In this scene, there is a tall chair on a wooden floor.
> >     The floor is brown and flat, giving the room a warm and natural feel.
> >     The walls are white and attached to the floor, creating a clean and spacious atmosphere.
> >     There are two other chairs and a rectangular wooden table on the floor.
> >     The chairs are tall and rigid, while the table is also tall and tidy.
> >     The color of both the chairs and the table is brown, matching the color of the floor.
> >     The chairs are positioned close to the table, with one chair in front of the other and the table in between them.
> >     This arrangement suggests that the chairs and table are being used for dining or socializing.
> >     Overall, the room appears to be a comfortable and stylish dining area.
> >      ```
> >      As you can see, most of the produced captions are correct in terms of objects, their attributes, and most importantly, their relations to other objects. Indeed, there is a minor error in the color of chairs and we commit to conducting a comprehensive check of our dataset. However, we do believe most of our data produced with the assistance of ChatGPT is accurate. Please find more examples (including scene graphs, our prompts, and ChatGPT responses) [here](https://docs.google.com/document/d/10KawUDUWWvSYwEVMDuwYMOYcwsdtxL97Teu87-5lZLs/edit?usp=sharing).
> >
> > In general, leveraging LLMs like ChatGPT to produce high-quality training data in NLP and multimodal ML has been a widely accepted idea. Some seminal examples include Self-Instruct [1], Alpaca [2], GPT-4-LLM [3], LLaVA [4], MIMIC-IT [5], LRV-Instruction [6], 3D-LLM [7]), we believe our scene-graph-based approach, which further extends to 3D scene-graph prompts and enhances the reliability, can serve as an efficient and robust pipeline to fill the blank in 3D VL data.
> >
> >
> > > 2. The ablations performed in the paper are unclear and inadequate. The paper doesn’t perform any component level ablations on the model...it’s unclear what the take-away message from ablations (Table 18, and Figure 4(a)) is.
> >
> > We thank you for your comments. We've conducted the blind experiments you mentioned and here are the results:
> >
> > | Tasks | Cap3D | SceneCap | Scan2Cap | ScanQA | SQA3D | 3RQA |
> > | --- | --- | --- | --- | --- | --- | --- |
> > | LEO-NoAct | 35.9 | 65.5 | 64.9 | 36.8 | 54.1 | 61.3 |
> > | LEO-NoAct-Blind | 16.2 | 58.6 | 43.4 | 32.8 | 50.7 | 51.5 |
> >
> > It can be seen that, without visual input, the model clearly cannot perform well on our 3D VL tasks. For the ablations without 2D and 3D input, please note that only the embodied action tasks (navigation and manipulation) require 2D input (please find this information in Table 1). Therefore, there is no need to perform without 2D ablation on other tasks. For the two embodied AI tasks, given the limited time, we've performed the ablation without 3D and without 2D with the navigation task, here are the results on the standard MP3D objnav validation (yes, this looks quite different from the original objnav experiment we did because we've reworked this part. Please take a look at the general response for more details):
> >
> > |  | MP3D-val |  |
> > | --- | --- | --- |
> > | Model | Success (↑) | SPL (↑) |
> > | Habitat-web (shortest) | 4.4 | 2.2 |
> > | Habitat-web (70k demo) | 35.4 | 10.2 |
> > | LEO (w/o 3D) | 8.6 | 6.8 |
> > | LEO (w/o 2D) | 7.8 | 4.6 |
> > | LEO (full) | 23.1 | 15.2 |

---

> ### Author Response · Authors · 2023-11-20
> **Response to Reviewer ADo8 (3/N)**
>
> As you can see, without 3D or 2D information both lead to significant performance drop. Some intuition: since LEO does not have a recurrent policy like RNN (effectively, LEO is similar to [RT-2](https://arxiv.org/abs/2307.15818) in terms of the transformer-based feedforward policy), therefore in the navigation task, 2D provide necessary information for avoiding obstacles like walls, while 3D provides global map details making it possible for a feed-forward policy in LEO to learn to navigate to the target in the shorest path. We will include a more comprehensive ablation on 2D/3D in the final version.
>
> For the existing ablations on data presented in Table 18 and Figure 4(a), our take-home messages have been made clear in sec. 4.4:
>
> - The two-stage training (align-then-instruct) is crucial, compared to instruction-tuning only.
> - Compositional generalization is challenging. The variant *ScanNetOnly*, after learning captioning on 3RScan (stage 1) and QA on ScanNet (stage 2), fails to generalize to perform QA on 3RScan.
> - More general data leads to weaker task-specific performances. Adding new-domain data slightly harms in-domain performances, e.g., Scan2Cap, (*ScanNetOnly* vs. *NoAct (VL)*). Scaling up instruction-tuning data brings significant improvements (*PartialData* vs. *Full (VLA)*). The incorporation of embodied acting data weakens VL performances (*Full (VLA)* vs. *NoAct (VL)*).
>
>
> > 3. ...This is an insufficient way to justify design decisions...I think relying on GPT-4 to subjectively evaluate two captions and checking whether it’s sufficiently grounded to a 3D scene is problematic.
>
> Thanks for raising this. The qualitative examples in our paper are used to illustrate the comparisons. The best way to rigorously assess the quality of such free-form data is human evaluation. Actually, we have manually examined numerous data examples to ensure the reliability of our data and meanwhile polish our refinement procedures. In Appendix A, we present quantitative analysis to justify our refinement procedures. To further quantitatively showcase our superiority over 3D-LLM [7], we analyze the answer accuracy for Object Counting questions based on the 3D-LLM released data and our data (it is hard to compare the data directly due to different source scenes). We present the results here:
>
> |  | 3D-LLM | ours w/o O-CoT | ours with O-CoT | ours after refinement |
> | --- | --- | --- | --- | --- |
> | accuracy (%) | 56.45 | 57.41 | 78.02 | 100 |
>
> Specifically, we consider the questions starting with “How many” and ending with “in the room/bedroom/kitchen”, and check the correctness according to the ground-truth labels/IDs. The accuracy of 3D-LLM released data is close to our raw accuracy (56.45 vs. 57.41), which indicates the similar performance of ChatGPT without advanced prompting technology such as O-CoT. This also shows the necessity of refinement procedures. Since there are no other salient response types such as Object Existence (e.g., “Is there a table in the room?”) in 3D-LLM’s data, we only consider the accuracy of Object Counting as a quantitative indicator.
>
> To sum up, we believe our data construction approach, which is indeed a combination of prompt ChatGPT through our proposed O-CoT and human examination, could provide accurate 3D-language data for training LEO. Moreover, our data quality is considerably better than our counterparts (3D-LLM).
>
> > ...it’s unclear if the GT can be trusted or not. This puts into question the entire GT dataset.
>
> Thanks for your comments. All the datasets for evaluating LEO (those appear in Table 2), are standard benchmarks introduced by the research community, not collected by us (and we believe they were not produced by ChatGPT). Therefore, the possible errors (which we believe are rare) in the constructed datasets, LEO-align and LEO-instruct should not affect the evaluation of LEO.

---

> ### Author Response · Authors · 2023-11-20
> **Response to Reviewer ADo8 (4/N)**
>
> > 4. Results on object-navigation are surprisingly low...Can the authors please clarify the experimental setup for object navigation?
>
> Thank you for pointing this out! We’ve spotted some major issues in our objnav evaluation protocol since the submission of our manuscript. Therefore, we’ve re-worked the whole objnav experiment and that part in both sec. 4.3 and the appendix has been completely renovated. Here are some key insights:
>
> 1. We’ve switched to using the standard habitat MP3D objnav validation set (MP3D-val) and the protocol available in habitat-lab (https://github.com/facebookresearch/habitat-lab). Therefore, comparing with baselines like Habitat-web becomes possible. Additionally, we also evaluate LEO on the newly introduced HM3D objnav validation set (HM3D-val) (note: this is “zero-shot” as only MP3D scenes are used for training LEO) and compare it with a strong baseline from cortexbench [9]. For your convenience, we’ve posted the results below.
>
>     |  | MP3D-val |  | HM3D-val |  |
>     | --- | --- | --- | --- | --- |
>     | Model | Success (↑) | SPL (↑) | Success (↑) | SPL (↑) |
>     | H.w. (shortest) | 4.4 | 2.2 | - | - |
>     | H.w. (70k demo) | 35.4 | 10.2 | - | - |
>     | VC-1 (ViT-B) | - | - | 57.1 | 31.4 |
>     | LEO | 23.1 | 15.2 | 23.1 | 19.1 |
>
>     We would like to summarize the findings below:
>
>     - First of all, after resolving this evaluation issue, LEO is in fact producing **reasonable** performances on both standard MP3D-val and even HM3D-val in a zero-shot fashion.
>     - Thanks to the 3D branch within LEO that processes 3D input and provides global 3D information(potentially offering coarse map details), LEO is able to attain better SPL than Habitat-web baseline, meaning that LEO is able to take shorter paths by leveraging the map information.
>     - Compared to VC-1, an agent trained on HM3D, LEO offers reasonable results given the fact that it has never been trained on HM3D scenes.
>
> 2. Per your concern on LEO trained with human demonstrations vs. shortest path, we provide the following results in the appendix:
>
>     |  | MP3D-seen |  | MP3D-unseen |  |
>     | --- | --- | --- | --- | --- |
>     | Model | Success (↑) | SPL (↑) | Success (↑) | SPL (↑) |
>     | Habitat-web (shortest) | 4.4 | 2.2 | - | - |
>     | Habitat-web (70k demo) | 35.4 | 10.2 | - | - |
>     | LEO (shortest) | 23.1 | 15.2 | 11.1 | 9.6 |
>     | LEO (70k demo) | 7.1 | 5.3 | 8.9 | 8.6 |
>
>     As you can see, training with human demonstrations indeed leads to much worse performances of LEO and we believe the root cause is intuitive:
>
>     - LEO is able to process the extra global 3D scene input (a list of objects in the scene, including their point cloud and bbox information), therefore, learning with shortest path data makes sense — LEO could learn to convert the 3D scene input into map details, then learn to navigate to the target in shortest path by learning from the shortest path navigation data.
>     - Further, as in our updated sec. 4.3 and H.2, LEO indeed **does not have any recurrent structure** when it comes to action prediction, i.e. the only thing relayed from the past is an action history of 4 — LEO effectively can be viewed as a transformer-based feed-forward policy, similar to RT-2 [8]. Therefore learning from human demonstration could be extra difficult, as it includes exploration, which can be hard to learn without a recurrent structure like RNN.
>     - Note that the choice of not having a recurrent structure is a result of the computation efficiency and scalability vs. performance tradeoffs. We commit to exploring more on this in future work.
>
> > How does the agent handle multi-step trajectory of the agent (observation action pair for multiple time steps)? Is the agent only trained as a markovian policy which doesn’t require encoding the history? Or is the trajectory encoded by simply appending each observations sequentially into the LLM?
>
> Thanks for your comments. As we mentioned in the previous response, LEO is indeed a Markovian policy without any recurrent structure, and the only thing relayed from the past are 4 past actions. Therefore, 3D information (potentially offering map details) is indeed crucial, and learning from human demonstrations could fail as such trajectories include exploration and can be hard to learn without a recurrent structure.
>
> > CLIPort numbers in Table 3 are lower than the highest reported numbers in the CLIPort paper. For completeness and accuracy, can the authors update the CLIPort numbers to the highest numbers reported in the paper?
>
> Thanks for raising this. We've checked the original CLIPort paper. We have added the performances of single-task CLIPort in Table 3 and discarded the "multi-attr" CLIPort variant, which utilizes "unseen" tasks (except for the task being evaluated). **This never happens in LEO** as only seen tasks are used for training. Therefore, comparing with these numbers would be **unfair** and we choose not to show them.

---

> ### Author Response · Authors · 2023-11-20
> **Response to Reviewer ADo8 (5/N)**
>
> We hope the above response can resolve your questions and concerns. Please let us know if there is any further question!
>
> [1] Yizhong Wang, et al. Self-instruct: Aligning language model with self generated instructions. ACL 2023.
>
> [2] Rohan Taori, et al. Stanford alpaca: An instruction-following llama model. https://github.com/tatsu-lab/stanford_alpaca, 2023.
>
> [3] Baolin Peng, et al. Instruction tuning with gpt-4. arXiv preprint arXiv:2304.03277, 2023.
>
> [4] Haotian Liu, et al. Visual instruction tuning. NeurIPS 2023.
>
> [5] Bo Li, et al. Mimic-it: Multi-modal in-context instruction tuning. arXiv preprint arXiv:2306.05425,
> 2023.
>
> [6] Fuxiao Liu, et al. Mitigating hallucination in large multi-modal models via robust instruction tuning. arXiv preprint arXiv:2306.14565, 2023.
>
> [7] Yining Hong, et al. 3d-llm: Injecting the 3d world into large language models. NeurIPS 2023.
>
> [8] Anthony Brohan, et al. Rt-2: Vision-language-action models transfer web knowledge to robotic control. arXiv preprint arXiv:2307.15818, 2023.
>
> [9] Arjun Majumdar, et al. Where are we in the search for an artificial visual cortex for embodied intelligence? arXiv preprint arXiv:2303.18240, 2023.

---

> ### Author Response · Authors · 2023-11-22
> **Do our responses address your concerns? Thank you again for your valuable feedback!**
>
> Dear Reviewer ADo8,
>
> We encourage you to view the demo video and animations we have provided at https://generalist-leo-anonymous.github.io. These resources offer a clearer understanding of the tasks we have benchmarked and the design of our model. We also added more visualizations of our collected data regarding at [this URL](https://docs.google.com/document/d/10KawUDUWWvSYwEVMDuwYMOYcwsdtxL97Teu87-5lZLs/edit?usp=sharing).
>
> **As we are approaching the end of the author-reviewer discussion phase, we want to check if our responses address your concerns well. Are there any further clarifications we could provide? We look forward to your feedback. Please feel free to let us know if we have missed any issues.**
>
> Thank you again for your valuable feedback!
>
> Best,
>
> Authors

---

### Author Response · Authors · 2023-11-20
**Thank you all for the time and the invaluable suggestions! Here are some general responses:**

We thank all reviewers for their insightful comments and acknowledgment of our contributions. We highlight the major contributions of our work as follows:


- We've attempted a fairly ambitious idea of building generalist embodied agents, which points a novel and useful research direction (Reviewer ADo8, foVx, bU9n), with not only 2D but also 3D input (Reviewer ADo8, foVx). The efforts of integrating multiple modalities and tasks are commendable (Reviewer ADo8, foVx, Vn42, bU9n)
- We've made substantial contributions to exploring the idea of constructing high-quality data with LLMs for training such agents (Reviewer ADo8, foVx).
- The implementation details, including model design and training, provide useful insights to the community (Reviewer foVx)
- The manuscript is clearly written and graphics are well-presented (Reviewer bU9n).


We’ve revised our manuscript per the reviewers’ suggestions (highlighted in red in the uploaded revision pdf). Detailed responses to each reviewer’s concerns are carefully addressed point-by-point. Below summarize the major updates we’ve made:

- Revised introduction (sec. 1): scalable models -> unified models, etc
- A **completely reworked navigation experiment**, with new results on the standard dataset(MP3D-val and HM3D-val), baseline comparison, OOD tests,  evaluation protocols, and model implementation details (sec. 4.3, table 4 and appendix H.2, I.1)
- Additional details on our prompting techniques for data collection (Figure 11-13)
- Renovated and additional explanation, statistics visualization, and examples of our two datasets: LEO-align and LEO-instruct (sec. C, table 24-26, figure 14-19)
- Additional numerical results on how different data schemes (no alignment, less data, no action data, domain-specific data) affect the performances (table 18)
- More examples from the testing dataset, ex. ScanQA, Scan2Cap, SQA3D, etc (table 19, table 21-23)
- Other fixes suggested by the reviewers.

We believe our model could make a timely contribution to both the 3D understanding and embodied AI community.  It is our pleasure to hear your feedback, and we look forward to answering your follow-up questions.

Best,

Authors

---

### Meta-Review · Area_Chair_1D4D · 2023-12-14

**Metareview:**

The authors present a very ambitious goal to achieve a so-called embodied generalist agent in 3D world. The proposed approach aims to generate both text as well as actions. The effort to integrate all these signals into the model, and setting up training on so many tasks is challenging, and I appreciate the authors' effort.

However, big claims need extra careful and strong data for validation. As pointed out by the reviewers, here are some points that could improve the work and draft: The LLM-based data collection raises concerns about accuracy due to grammatical errors, misinterpretations of scenes, and potential biases in the underlying datasets. This casts doubt on the reliability of the entire dataset and subsequent results. The proposed approach primarily combines existing techniques (tokenization, embedding, LLM, training & inference). While the multi-task learning aspect is interesting, the paper still lacks a clear focus on specific tasks. Results on object navigation from the original draft are actually low compared to other methods. Missing baselines and lack of comparison to existing work weaken the evaluation of the agent's capabilities.

I would also suggest the authors refrain from naming an agent a specific name like LEO. It is misleading and confusing and overshadows the actual scientific questions that you are actually trying to address.

**Justification For Why Not Higher Score:**

The paper aims to build a "generalist agent" but I agree with the concerns raised by reviewers, it sort-of fails to clearly define its scope or demonstrate its progress towards general intelligence. The connection between 3D understanding and achieving general intelligence is not well-explained and established, and lacking strong data to support this ambitious argument.

**Justification For Why Not Lower Score:**

N/A

---

### Decision · Program_Chairs · 2024-01-16

Reject